# Scalable Policy-Based RL Algorithms for POMDPs

**Ameya Anjarlekar**
UIUC
ameyasa2@illinois.edu

**S. Rasoul Etesami**
UIUC
etesami1@illinois.edu

**R. Srikant**
UIUC
rsrikant@illinois.edu

## Abstract

The continuous nature of belief states in POMDPs presents significant computational challenges in learning the optimal policy. In this paper, we consider an approach that solves a Partially Observable Reinforcement Learning (PORL) problem by approximating the corresponding POMDP model into a finite-state Markov Decision Process (MDP) (called *Superstate* MDP). We first derive theoretical guarantees that improve upon prior work that relate the optimal value function of the transformed Superstate MDP to the optimal value function of the original POMDP. Next, we propose a policy-based learning approach with linear function approximation to learn the optimal policy for the *Superstate* MDP. Consequently, our approach shows that a POMDP can be approximately solved using TD-learning followed by Policy Optimization by treating it as an MDP, where the MDP state corresponds to a finite history. We show that the approximation error decreases exponentially with the length of this history. To the best of our knowledge, our finite-time bounds are the first to explicitly quantify the error introduced when applying standard TD learning to a setting where the true dynamics are not Markovian.

## 1 Introduction

Reinforcement learning (RL) provides a robust and systematic framework for solving sequential decision-making problems by modeling these problems as Markov Decision Processes (MDPs). However, many real-world systems such as robotic controllers must operate under uncertainty, handling incomplete, noisy, or ambiguous data, making traditional RL techniques ineffective for such problems Cassandra et al. [1996].

To address such challenges, partially observable reinforcement learning (PORL) extends the RL framework by modeling these problems as Partially Observable Markov Decision Processes (POMDPs). This approach accounts for hidden states, enabling effective decision-making under uncertainty. Beyond these applications, PORL is widely applied in areas such as autonomous driving Levinson et al. [2011], personalized content recommendations Li et al. [2010], medical diagnosis Hauskrecht and Fraser [2000], and games Brown and Sandholm [2019], where decision-making under partial observability is crucial. As a result, addressing decision-making under uncertainty has become a critical topic across diverse fields such as operations research and healthcare.

Although POMDPs provide a general framework for modeling decision-making under uncertainty, solving them presents significant computational challenges even when the exact model of the POMDP is known. The early works of Smallwood and Sondik [1973], Åström [1965] focused on the planning problem and effectively converted POMDPs into MDPs via belief states. However, the continuous

39th Conference on Neural Information Processing Systems (NeurIPS 2025).

nature of belief states and the PSPACE-completeness of solving POMDPs Papadimitriou and Tsitsiklis [1999], Vlassis et al. [2012] make these problems computationally intractable.

Various approaches have been proposed to address these limitations, with the aim of providing approximate solutions for learning POMDPs by selecting actions based on a finite history of observations. For example, Jaakkola et al. [1994], Williams and Singh [1998], Azizzadenesheli et al. [2016] select actions based solely on current observations, while others, such as Loch and Singh [1998], Littman [1994], consider past $k$ observations to guide decision making. Nevertheless, while these approaches demonstrate empirical success, they lack rigorous theoretical performance guarantees.

In contrast, for learning fully observable MDPs, a wide range of Reinforcement Learning algorithms exist with provable sample complexity bounds, ranging from value-iteration-based approaches such as Q-learning to policy-iteration-based algorithms such as actor-critic and natural policy gradient (NPG). A comprehensive survey of these algorithms can be found in Bertsekas and Tsitsiklis [1996] and Sutton and Barto [2018].

The empirical success of finite-history-based algorithms for POMDPs and the availability of performance guarantees for standard RL algorithms motivate a critical research question: *Can we leverage standard RL algorithms to approximately learn an optimal policy for a POMDP by treating it as an MDP, where the states correspond to the finite histories? Specifically, can we establish theoretical performance bounds for such an approach?*

In this paper, we address this question by considering a previously proposed approximation by Kara and Yüksel [2023] that maps a POMDP to a finite-state MDP, called the *super-state* MDP by restricting the information at each time instant to a finite history of past observations. However, this approximation is difficult to study for the following reasons:

- It is unclear whether using standard TD learning as the policy evaluation step in a problem where the true model is a POMDP would result in good performance or even convergence.
- The TD learning algorithm used in Cayci et al. [2024] uses a $m$-step version of TD as a workaround, which is computationally more expensive than standard TD learning.

In view of the limitations of the prior work, our main contributions are stated as follows:

- **Approximation guarantees for POMDP to MDP transformation:** A standard approach to assess the quality of a transformed MDP is to bound the difference between its optimal value function and that of the original POMDP. In prior work such as Kara and Yüksel [2023], Subramanian and Mahajan [2019], this difference is bounded by the square of the expected horizon length, i.e., $\frac{1}{(1-\gamma)^2}$ where $\gamma$ is the discount factor, while, Abel et al. [2016] provides a bound of $\frac{1}{(1-\gamma)^3}$, and the bound in Cayci et al. [2024] is polynomial in the horizon length. We improve upon these results by providing a tighter bound. Additionally, while Kara and Yüksel [2023] provides bounds on the expected difference between the optimal value functions, we establish a stronger worst-case bound.

- **A General Purpose Algebraic Identity**: A key reason for improved bounds is the introduction of a novel algebraic result, i.e., Lemma 2 which proves effective in bounding expressions of the form $\left|\sum_{i=1}^{m}(a_i b_i - c_i d_i)\right|$, when vectors $(\boldsymbol{a}, \boldsymbol{c})$ and $(\boldsymbol{b}, \boldsymbol{d})$ are close to each other, and $(\boldsymbol{b}, \boldsymbol{d})$ corresponds to probability mass functions. Traditional approaches rely on decompositions and triangle inequality, often leading to loose bounds. In contrast, our refined approach yields tighter guarantees, improving the existing bounds.

  Our algebraic result is of broader interest and can be used to improve bounds in other analyses. As an example, we refine[1] the performance bounds in Subramanian and Mahajan [2019], which introduced approximate information states but lacked a systematic method for constructing them. Additionally, while their deep learning-based approach provides an empirical solution, it optimizes an objective misaligned with the theoretical guarantees.

- **Addressing Challenges in Policy Optimization:** Having established that the Superstate MDP is a strong approximation of the POMDP, we investigate whether standard Policy Optimization algorithms can effectively learn its optimal policy. A key challenge arises from the fact that learning samples correspond to belief states of the original POMDP rather

---

[1]Proof is in the Appendix.

than the Superstate MDP, creating a sampling mismatch that raises concerns about whether applying standard RL techniques can yield strong performance guarantees.

Cayci et al. [2024] addresses this issue by modifying the TD-learning part of Policy Optimization by employing an $m$-step TD-learning approach, leading to significantly higher computational complexity. In contrast, our work is the first to establish convergence guarantees for a standard policy optimization algorithm that alternates between learning the Q-function for a fixed policy using TD-learning and updating the policy accordingly. This avoids additional computational overhead while providing stronger theoretical guarantees.

- **Performance Bounds for Linear Function Approximation Setting:** Apart from Cayci et al. [2024] which leads to a higher computational cost, Kara and Yüksel [2023] considers Q-learning to learn the optimal policy for the Superstate MDP. However, along with relying on ergodicity assumptions, their approach faces significant scalability challenges due to the difficulty of proving convergence for Q-learning with linear function approximation. In contrast, we develop performance bounds for the Policy Optimization Algorithm and extend our analysis to the function approximation case, enabling scalability to large state spaces.

  A key challenge in proving convergence for Policy Optimization algorithms lies in the non-stationary nature of the sampling policy during the TD-learning phase. In contrast, the Q-learning approach in Kara and Yüksel [2023] can leverage a stationary exploratory sampling policy, simplifying the analysis. Additionally, they assume an additional ergodicity condition and provide only asymptotic convergence guarantees. In comparison, our approach avoids such restrictive assumptions and establishes finite-time performance bounds.

  Finally, another minor contribution of our work is to extend the analysis for the POLITEX algorithm in Abbasi-Yadkori et al. [2019] to the discounted reward setting and analyze the regret of our algorithm with respect to the optimal value function of the POMDP.

## 2 Related Work

**Planning algorithms for POMDP:** Early POMDP algorithms assumed full model knowledge and focused on exact belief-state calculations. Techniques such as Witness algorithm Cassandra et al. [1994], Lovejoy's suboptimal algorithm Lovejoy [1993], and incremental pruning Zhang and Liu [1996] were aimed at making planning more efficient. Comprehensive surveys can be found in Krishnamurthy [2016], Murphy [2007]. However, these approaches suffer from exponential complexity growth due to reliance on precise belief-state calculations, limiting their practicality.

**Using internal state representations to solve PORL problems:** To address the challenge of continuous belief states, internal state representations have been proposed. Works like Jaakkola et al. [1994], Williams and Singh [1998], Azizzadenesheli et al. [2016] select actions based solely on current observations, while Loch and Singh [1998], Littman [1994], consider past $k$ observations to guide decision-making. Additionally, sliding window controllers Williams and Singh [1998], Loch and Singh [1998], Kara and Yüksel [2023], Sung et al. [2017], Yu [2005], Cayci and Eryilmaz [2024], Amiri and Magnússon [2024] and memory-based techniques McCallum [1993], Meuleau et al. [1999] have demonstrated practical success. However, most of these methods are heuristic and lack formal guarantees.

**Learning algorithms with provable guarantees:** Recent efforts have focused on developing algorithms for solving PORL problems with theoretical guarantees. Du et al. [2019], Efroni et al. [2022] provide provable bounds, though these rely on the state decodability assumption, which implies that a finite history of observations can perfectly infer the current state. Other approaches Wang et al. [2022], Liu et al. [2022], Jin et al. [2020] target specific subclasses of PORL problems.

**Deep learning inspired techniques:** It is also worth mentioning prior work on empirical approaches using deep learning. In particular, recurrent neural networks (RNNs) Hausknecht and Stone [2015], Wierstra et al. [2007] and variational encoders Igl et al. [2018] have been used to model the uncertainty in PORL by capturing temporal dependencies. While these techniques have demonstrated impressive empirical results, they lack rigorous performance guarantees.

## 3 Problem Description and Prior Results

In this section, we introduce the problem formulation and review some well-known prior results; for basic POMDP theory, the reader is referred to Krishnamurthy [2016].

### 3.1 POMDP Structure and Notations

We consider a general finite-state POMDP model, which can be characterized as follows:

- Consider a set of finite states $S$, with the state of the system at time $t$ denoted by $s_t \in \mathcal{S}$, which is *not* observed. Also, assume that the initial state $s_0$ is sampled from a distribution $\mathcal{D}$, with the corresponding probability mass function represented by the vector $\boldsymbol{\pi}_0$.

- At any time $t$, the agent chooses an action $a_t = a$ from a finite set of possible actions $\mathcal{A}$, where, for simplicity, we assume that all actions from $\mathcal{A}$ are feasible for every state. The state then evolves according to a transition probability

$$\mathcal{P}(s' \mid s, a) = \mathbb{P}(s_{t+1} = s' \mid s_t = s, a_t = a).$$

  Moreover, the agent receives a reward $r(s, a)$, which is the reward obtained if performing an action $a$ in state $s$, where we assume finite reward, i.e., there exists $\bar{r}$ such that $|r(s, a)| \leq \bar{r} \ \forall s, a$.

- Since states cannot be observed, we assume that information about them at each time $t$ is obtained through a noisy discrete observation channel whose output $y_t \in \mathcal{Y}$ is chosen according to a conditional distribution $\Phi(y \mid s) := \mathbb{P}(y_t = y \mid s_t = s)$, where $\mathcal{Y}$ is a finite set of observations. Therefore, after agent takes action $a_t$, it receives a reward $r_t$ and observes $y_{t+1}$.

- Let $H_t := \{a_0, y_1, a_1, \ldots, a_{t-1}, y_t\} \in \mathbb{H}$ be the entire observed history up to time $t$, where $\mathbb{H}$ refers to the set of all possible histories (of any length) of the POMDP. Given a history $H \in \mathbb{H}$ and an initial distribution $\boldsymbol{\pi}_0$ over the states, we denote the probability of being in state $s$ by $\pi(s \mid H) = \mathbb{P}(s \mid H, \boldsymbol{\pi}_0)$ and define the *belief state* given history $H$ by $\boldsymbol{\pi}(H) = (\pi(s \mid H))_{s \in \mathcal{S}}$. In particular, the belief state[2] $\boldsymbol{\pi}$ belongs to $\mathbb{B} \subseteq \Sigma(\mathcal{S})$, where $\Sigma(\mathcal{S}) := \{x \in \mathbb{R}_+^{|\mathcal{S}|} : \sum_{i=1}^{|\mathcal{S}|} x_i = 1\}$, and $\mathbb{B}$ refers to the set of all possible belief states that can be realized using a history $H \in \mathbb{H}$. Now, if at time $t$, the agent takes an action $a_t = a$ and observes $y_t = y$, by Bayes' rule, the belief state can be updated as follows

$$\pi(s \mid H_t) = \frac{\sum_{s'} \pi(s' \mid H_{t-1})\mathcal{P}(s \mid s', a)\Phi(y \mid s)}{\sum_{s''} \sum_{s'} \pi(s' \mid H_{t-1})\mathcal{P}(s'' \mid s', a)\Phi(y \mid s'')}, \tag{1}$$

  where $H_{t-1} = H_t \setminus \{y_t, a_t\}$. Therefore, any belief state $\boldsymbol{\pi}(H)$ can be calculated recursively from the history $H$ and using the belief update rule (1) with the belief at $t = 0$ given by $\boldsymbol{\pi}_0$.

- The agent's goal is to learn the optimal policy (defined more precisely in the next subsection) through sequential interactions with the environment.

### 3.2 Belief State MDP Representation of the POMDP

We note that a POMDP can be reduced to a fully observed MDP by considering the belief states as the states of the MDP. This is discussed in detail below:

- For any belief state $\boldsymbol{\pi}$, let $r(\boldsymbol{\pi}, a) := \sum_s \pi(s)r(s, a)$. Thus, the POMDP reduces to a fully observed MDP where $(\boldsymbol{\pi}, a) \in \mathbb{B} \times \mathcal{A}$ represents a state-action pair and $r(\boldsymbol{\pi}, a)$ is the corresponding reward for that state-action pair with the state transition law given by Eq. (1).

- For any belief state $\boldsymbol{\pi} \in \mathbb{B}$ and any policy $\mu$, we define the value function $V^\mu(\boldsymbol{\pi})$ as

$$V^\mu(\boldsymbol{\pi}) := \mathbb{E}\Big[ \sum_{t=0}^\infty \gamma^t r(\boldsymbol{\pi}_t, a_t) \mid \boldsymbol{\pi}_0 = \boldsymbol{\pi} \Big], \tag{2}$$

  where the belief states $\boldsymbol{\pi}_t := \boldsymbol{\pi}(H_t)$ evolve using the update rule (1), when the actions are taken according to $a_t \sim \mu(a \mid \boldsymbol{\pi}_t) \ \forall t \geq 0$ with the initial belief state $\boldsymbol{\pi}_0 = \boldsymbol{\pi}$. Here,

---

[2]When there is no ambiguity, we drop the dependency of belief state on the history and write $\boldsymbol{\pi} = \boldsymbol{\pi}(H)$.

$\gamma \in [0, 1)$ is a discount factor. Our goal is to find an optimal policy $\mu^*(a \mid \boldsymbol{\pi})$ corresponding to each belief state $\boldsymbol{\pi}$ to maximize the expected cumulative discounted rewards $V^\mu(\boldsymbol{\pi})$.

- For any belief state $\boldsymbol{\pi} \in \mathbb{B}$ and for a policy $\mu$, we define the Q-value function as

$$Q^\mu(\boldsymbol{\pi}, a) := \mathbb{E}\Big[ \sum_{t=0}^{\infty} \gamma^t r(\boldsymbol{\pi}_t, a_t) \mid \boldsymbol{\pi}_0 = \boldsymbol{\pi}, a_0 = a \Big],$$

  where the belief states $\boldsymbol{\pi}_t$ evolve using the update rule (1) and the actions taken according to $a_t \sim \mu(a \mid \boldsymbol{\pi}_t) \; \forall t \geq 1$.

### 3.3 Bellman's Optimality and Uniqueness of Solution

Here, we first present a result from Krishnamurthy [2016] showing that the optimal value function satisfies *Bellman's Optimality Equation*, which can also be used to establish the existence and uniqueness of an optimal solution to the POMDP.

**Theorem 1** *Consider an infinite horizon discounted reward POMDP with discount factor $\gamma \in [0, 1)$, and finite state and action spaces $\mathcal{S}$ and $\mathcal{A}$, respectively. Then,*

1. *For any belief state $\boldsymbol{\pi} \in \mathbb{B}$, the optimal expected cumulative discounted reward $\max_\mu V^\mu(\boldsymbol{\pi})$ is achieved by a stationary deterministic Markovian policy $\mu^*$.*

2. *For any belief state $\boldsymbol{\pi}(H) \in \mathbb{B}$, the optimal policy $\mu^*$ and the optimal value function $V^{\mu^*}$ satisfy the* Bellman's Optimality Equation: [3]

$$V^{\mu^*}\big(\boldsymbol{\pi}(H)\big) = \max_{a \in \mathcal{A}} \Big[ \sum_{s \in \mathcal{S}} \boldsymbol{\pi}(s|H) r(s, a) + \gamma \sum_{y \in \mathcal{Y}} V^{\mu^*}\big(\boldsymbol{\pi}(H \parallel \{y, a\})\big) \sigma(\boldsymbol{\pi}(H), y, a) \Big],$$

(3)

   *where $\sigma(\boldsymbol{\pi}(H), y, a)$ denotes the probability of the observation being $y$ conditioned on the previous belief state being $\boldsymbol{\pi}(H)$ and action $a$ is taken, which can be calculated explicitly as $\sigma(\boldsymbol{\pi}(H), y, a) = \sum_{s,s'} \Phi(y \mid s') \mathcal{P}(s' \mid s, a) \pi(s|H)$.*

3. *There always exists a unique solution to the Bellman Optimality Equation* (3).

Let us denote the Bellman optimality operator in (3) by $T(\cdot)$. Also, let $\mu^*$ be the optimal policy and $V^* = V^{\mu^*}$ be the optimal value function. Then, the optimal value function $V^*$ satisfies the fixed-point equation

$$V^* = T(V^*).$$

Therefore, using Theorem 1, for any belief state $\boldsymbol{\pi} = \boldsymbol{\pi}(H) \in \mathbb{B}$, we can write

$$V^*\big(\boldsymbol{\pi}\big) = \max_{a \in \mathcal{A}} \Big[ r(\boldsymbol{\pi}, a) + \gamma \sum_{y \in \mathcal{Y}} V^*\big(\boldsymbol{\pi}(H \parallel \{a, y\})\big) \sigma(\boldsymbol{\pi}, y, a) \Big].$$

Note that reducing the POMDP to an MDP using the belief-state formulation does not solve the problem of finding the optimal policy using reinforcement learning. This is because the belief states depend on the history, whose length increases with each time step. In the following section, we present a scalable approach to bypass this intractability in solving infinite horizon POMDPs.

## 4 Algorithm and Theoretical Results

In this section, we describe our solution approach, develop an algorithm for approximately solving POMDPs, and provide theoretical guarantees on the performance of the algorithm.

---

[3]The symbol $\parallel$ denotes cocatenation. For example if $H = \{y_0, a_0, y_1, a_1\}$, then, $H \parallel \{y, a\} = \{y_0, a_0, y_1, a_1, y, a\}$

## 4.1 Defining an Approximate MDP Using Finite Observations

**Definition 1** *Given a POMDP and a fixed constant $l \in \mathbb{N}$, let $\mathbb{H}_{\leq l}$ be the set of all histories of length at most $l$. We refer to any element of $\mathbb{H}_{\leq l}$ as a Superstate. Moreover, we define $\mathcal{G} : \mathbb{H} \to \mathbb{H}_{\leq l}$ as the grouping operator if for any finite-length history $H \in \mathbb{H}$, it returns the superstate obtained by truncating $H$ to its last $l$ action-observation elements. In particular, if $H_t = \{a_0, y_1, a_1, \ldots, a_{t-1}, y_t\}$, then[4]*

$$\mathcal{G}(H_t) = \{y_{\max\{1, t-l+1\}:t}, a_{\max\{0, t-l\}:t-1}\}, \quad \forall\, t \geq 1.$$

Next, we construct an MDP which consists of states corresponding to the Superstates. Since the number of Superstates is finite, our resulting Superstate MDP is a finite-state MDP, which can then be solved using existing RL techniques for finite-state MDPs in the literature. The Superstate MDP is defined as follows. For every pair of Superstates $B, B' \in \mathbb{H}_{\leq l}$, define

$$\tilde{r}(B, a) = \sum_s \pi(s \mid B) \cdot r(s, a), \tag{4}$$

$$\tilde{\mathcal{P}}(B' \mid B, a) = \sum_{y, s, s'} \mathbb{I}[\mathcal{G}(B \parallel \{y, a\}) = B'] \cdot \Phi(y \mid s') \mathcal{P}(s' \mid s, a) \pi(s \mid B), \tag{5}$$

where $\mathbb{I}[\cdot]$ is the logic indicator function and $B \parallel \{y, a\}$ means that the action-observation pair $\{y, a\}$ is concatenated to the end of the superstate $B$. Since the action and state spaces are bounded, an optimal value function always exists for a discounted MDP with bounded rewards. Let $\tilde{V}(\cdot)$ be the optimal value function for the Superstate MDP, which satisfies the following fixed-point equation

$$\tilde{V} = \tilde{T}(\tilde{V}),$$

where $\tilde{T}$ is the Bellman's optimality operator for the corresponding Superstate MDP. Therefore, for any superstate $B \in \mathbb{H}_{\leq l}$, we have

$$\tilde{T}(\tilde{V}(B)) = \max_{a \in \mathcal{A}} \left[ \tilde{r}(B, a) + \gamma \sum_{B'} \tilde{\mathcal{P}}(B' \mid B, a) \tilde{V}(B') \right].$$

Now, for a policy $\mu$ that is stationary with respect to the superstates, let us define the Bellman's operator $\tilde{T}^\mu$ by

$$\tilde{T}^\mu(V)(B) = \mathbb{E}_{a \sim \mu(\cdot \mid B)} \left[ \tilde{r}(B, a) + \gamma \sum_{B'} \tilde{\mathcal{P}}(B' \mid B, a) V(B') \right] \quad \forall B \in \mathbb{H}_{\leq l}.$$

Then, the value function of the Superstate MDP with respect to policy $\mu$, denoted by $\tilde{V}^\mu$, must satisfy the fixed-point Bellman's operator, i.e.,

$$\tilde{V}^\mu = \tilde{T}^\mu(\tilde{V}^\mu).$$

## 4.2 How good is the approximate MDP?

In this section, we compare optimal value function of the POMDP, denoted by $V^*$, with optimal value function of the Superstate MDP, denoted by $\tilde{V}$. We will show that error between these two optimal value functions decays exponentially with the length of the truncated history $l$. To that end, we first state a standard assumption in filtering theory, i.e., the *Uniform Filter Stability Condition*, also used in prior work van Handel [2008], which would be needed for our analysis to relate $V^*$ and $\tilde{V}$. This condition ensures sufficient mixing, preventing the system from being trapped in a subset of states.

**Assumption 1** *[Uniform Filter Stability Condition] Given any $\pi, \pi' \in \Sigma(\mathcal{S})$, and any $a, y$, let $K_{a,y}$ be an operator such that*

$$(K_{a,y} \otimes v)(s) = \frac{\sum_{s'} v(s') \mathcal{P}(s \mid s', a) \Phi(y \mid s)}{\sum_{s''} \sum_{s'} v(s') \mathcal{P}(s'' \mid s', a) \Phi(y \mid s'')}$$

*Then, there exists $\rho \in (0, 1)$ such that for any $\pi, \pi' \in \Sigma(\mathcal{S})$, we have[5]*

$$\|K_{a,y} \otimes \pi - K_{a,y} \otimes \pi'\|_{TV} \leq (1 - \rho) \cdot \|\pi - \pi'\|_{TV} \quad \forall a \in \mathcal{A}, y \in \mathcal{Y}.$$

---

[4]By notation $y_{r:t} = \{y_r, y_{r+1}, \ldots, y_t\}$ and $a_{r:t} = \{a_r, a_{r+1}, \ldots, a_t\}$ for any integers $r \leq t$.

[5]$\|\cdot\|_{TV}$ is the total variation norm

### 4.3 Intuition about Assumption 1

Assumption 1, i.e., the Uniform Filter Stability Condition means that every new observation sufficiently informs the agent to reduce differences between any two prior beliefs — ensuring the belief state "forgets" initial uncertainty and the filtering process is well-behaved.

Now to ensure that the filter stability condition holds we need

1) **Sufficiently Mixing State Transitions**: The transition kernel $\mathcal{P}(s' \mid s, a)$ should be mixing, meaning from any state s, there's a positive probability to reach many other states s' over time.

2) **Non-deterministic and Informative Observations**: The observation kernel $\Phi(y \mid s)$ must be non-deterministic but sufficiently informative, i.e., observations must have some noise or randomness.

To check whether the filter stability condition holds, one sufficient condition uses the Dobrushin Coefficients of the Transition kernel and the Observation Kernel.

From Theorem 5 of Kara and Yuksel [2020], if $(1 - \delta(\mathcal{P}))(1 - \delta(\Phi)) < 1$, then the filter stability condition holds. Here $\delta(\mathcal{P})$ is the minimum Dobrushin coeffient of the Probability Transition kernel across all possible actions and $\delta(\Phi)$ is the Dobrushin coefficient of the Observation kernel.

For example, the Dobrushin coefficient for the Probability Transition matrix $\delta(\mathcal{P})$ is defined as:

$$\delta(\mathcal{P}) = \inf_a \left[ \inf_{x,y \in \mathbb{S}} \sum_{z \in \mathbb{S}} \min \left( \mathcal{P}(z \mid x, a), \mathcal{P}(z \mid y, a) \right) \right]$$

The term inside $\inf_a$ quantifies the minimum overlap between any two rows of the matrix $\mathcal{P}(. \mid ., a)$.

Therefore, this quantity measures how similar the rows of the matrix are; smaller values indicate more mixing and hence, stronger contraction properties. As a result, when there is sufficient mixing—i.e., when every state has a non-trivial probability of transitioning to several other states or when every observation can be generated from multiple underlying states—the system exhibits filter stability.

This scenario is common in highly noisy or dynamic environments, where the belief update is less sensitive to the prior and more influenced by new observations. We also present in the Appendix B.1 a model for a practical example where Assumption 1 holds.

Additionally, for applications that do not satisfy Assumption 1, one could consider a multi-step variant where the system exhibits contraction after every $k$ steps. We believe our results could be extended under this milder assumption, making it a promising direction for future work.

Now, using this Uniform Filter Stability Condition, we prove Lemma 1 [6], which shows that all belief states corresponding to the same Superstate are close to each other in terms of total variation distance. Further, the distance decays exponentially fast with the length of the truncated history.

**Lemma 1** *Let $H$ and $H'$ be two different histories corresponding to the same superstate, i.e., $\mathcal{G}(H) = \mathcal{G}(H')$. Then, under Assumption 1, we have*

$$\left\| \boldsymbol{\pi}(H) - \boldsymbol{\pi}(H') \right\|_{TV} \leq (1 - \rho)^l. \tag{6}$$

If two histories belong to the same superstate, their sequences of (action, observation) pairs over the past $l$ steps are identical. Lemma 1 leverages the intuition that sufficiently informative observations allow the recent history to capture the current system state. Consequently, starting from different initial beliefs, the combination of informative observations and strong mixing ensures that, after enough time, the resulting belief states converge to similar distributions.

Next, we establish an algebraic inequality that will be used to obtain tighter optimality bounds.

**Lemma 2** *Let $\boldsymbol{a}, \boldsymbol{b}, \boldsymbol{c}, \boldsymbol{d} \in \mathbb{R}_+^m$ be positive vectors such that $\sum_{i=1}^m a_i = \sum_{i=1}^m c_i = 1$. Then,*

$$\left| \sum_{i=1}^m a_i b_i - \sum_{i=1}^m c_i d_i \right| \leq \frac{\|a - c\|_1}{2} \max(\|b\|_\infty, \|d\|_\infty) + \|b - d\|_\infty - \frac{\|a - c\|_1}{4} \|b - d\|_\infty. \tag{7}$$

Finally, using the above lemmas, we can state one of our main results.

---

[6]Proof of this lemma, as well as all subsequent lemmas and theorems, can be found in the Appendix

**Theorem 2** *Let $V^*$ be the optimal value function corresponding to the POMDP (or the belief state MDP), and let $\tilde{V}$ be the optimal value function corresponding to the Superstate MDP. Then, under Assumption 1 for every history $H \in \mathbb{H}$ with the corresponding belief state $\boldsymbol{\pi}(H) \in \mathbb{B}$ and the superstate $\mathcal{G}(H) \in \mathbb{H}_{\leq l}$, we have*

$$\left\| V^*\big(\boldsymbol{\pi}(H)\big) - \tilde{V}\big(\mathcal{G}(H)\big) \right\|_\infty \leq \frac{2\bar{r}(1-\rho)^l}{1-\gamma} + \frac{2\bar{r}\gamma(1-\rho)^l}{(1-\gamma)\big((1-\gamma)+\gamma(1-\rho)^l\big)} := \xi_{POMDP}^{SMDP}. \quad (8)$$

Thus, the above result relates the optimal value functions of the POMDP and the Superstate MDP. Note that the difference between the two value functions decreases exponentially with the length of the truncated history $l$, with the error effectively becoming 0 as $l \to \infty$. Next, we propose an algorithm to learn the optimal policy corresponding to the Superstate MDP.

### 4.4 An Approximate Policy Optimization Algorithm to Learn the Optimal Policy corresponding to the Superstate MDP

To learn the optimal value function for the Superstate MDP, one might consider standard reinforcement learning techniques such as Policy Optimization, which involves alternately learning the Q-function for a fixed policy and updating the policy. However, this learning process is not straightforward because the samples obtained at any time $t$ correspond to the actual belief state $\boldsymbol{\pi}(H_t)$, rather than the Superstate $\mathcal{G}(H_t)$. These issues due to sampling mismatch make the analysis of the TD-learning part of the algorithm non-trivial.

Additionally, we also consider the linear function approximation setting, where, given feature set $\Phi = \{\phi(B,a) \in \mathbb{R}^d : B \in \mathbb{H}_{\leq l}, \ a \in \mathcal{A}\}$, we aim to find the best $\theta \in \mathcal{B}(R)$[7] for some $R > 0$, such that $Q(B,a) = \phi^T(B,a)\theta$.[8] Further, since the feature vectors are bounded, we assume, without loss of generality, that $\|\phi(B,a)\|_2 \leq 1$. Note that for the function approximation, we constrain the parameters $\theta$ to lie within a ball of finite radius $R$. This is done to simplify the analysis of the function approximation part of the algorithm. The case without such a projection can be analyzed as in Srikant and Ying [2019], Mitra [2024].

---

**Algorithm 1: An Approximate TD Learning Algorithm for Superstate MDP**

**Input:** A fixed policy $\mu(\cdot \mid B)$, which is stationary with respect to the Superstates $B \in \mathbb{H}_{\leq l}$, discount factor $\gamma$, projection radius $R > 0$, stepsize sequence $\{\epsilon_t\}$, total iterations $\tau + l'$
Initialize $\theta_l$ randomly in $\mathcal{B}(R)$
Sample $s_0 \sim \mathcal{D}$ and set $H_0 = \{\}$
**for** $t = 0, 1, \ldots, \tau + l' - 1$ **do**
    Select action $a_t$ according to the policy $\mu(\cdot \mid \mathcal{G}(H_t))$
    Receive reward $r_t$ and observe $y_{t+1}$
    Update the history $H_{t+1} = H_t \| \{a_t, y_{t+1}\}$
    Select action $a_{t+1}$ according to the policy $\mu(\cdot \mid \mathcal{G}(H_{t+1}))$
    **if** $t \geq l'$ **then**
        $\theta_{t+1/2} = \theta_t + \epsilon_t\big[r_t + \gamma\phi^T\big(\mathcal{G}(H_{t+1}), a_{t+1}\big)\theta_t - \phi^T\big(\mathcal{G}(H_t), a_t\big)\theta_t\big] \cdot \phi(\mathcal{G}(H_t), a_t)$
        $\theta_{t+1} = \text{Proj}_{\mathcal{B}(R)}(\theta_{t+1/2})$
    **end**
**end**
**Output:** $\bar{Q}^\pi(B,a) = \Phi^T(B,a)\theta_{\tau+l'}$.

---

We now use a standard Temporal Difference (TD) learning algorithm to learn the Q-function of the Superstate MDP corresponding to a fixed policy $\mu$. The main idea is to perform a TD-update at every time $t$, to the value function of $\mathcal{G}(H_t)$ using the reward $r_t$ and subsequent observation $y_{t+1}$. This process is summarized in Algorithm 1. While the TD learning algorithm is standard, we note that the model to which the algorithm is applied is not standard: we apply the algorithm pretending that the underlying model is an MDP while the true model is a POMDP. We leverage a key insight: if two belief states are close in total variation distance, their reward and transition functions can be proved

---

[7]$\mathcal{B}(R)$ denotes a $d$-dimensional Euclidean ball of radius $R$.
[8]Here, the superscript $T$ refers to the transpose of the vector $\phi$.

to be close, allowing us to perform TD-learning by pretending that the underlying model is Superstate MDP while the true model is actually a POMDP.

Next, we show that under Assumption 1 and with sufficient exploration by the policies, the Superstate MDP admits a contraction mapping.

**Lemma 3** *Let $\tilde{\mathcal{P}}^\mu \in \mathbb{R}_+^{|\mathbb{H}_{\le l}| \times |\mathbb{H}_{\le l}| \times |\mathcal{A}|}$ be the probability transition matrix defined by Eq. (5) and following policy $\mu$. Then, there exists a constant $\rho' \in (0,1)$ such that for all pairs of distributions $d_1, d_2$ over the superstates such that $(d_1 - d_2)(i) \ne 0 \, \forall \, i$ and for all policies with sufficient exploration $\mu(a \mid B) \ge \delta \, \forall \, a, B$, such that $(1-\rho)^l < \delta |\mathbb{A}|$ we have*

$$\left\| \tilde{\mathcal{P}}^\mu d_1 - \tilde{\mathcal{P}}^\mu d_2 \right\|_{TV} \le (1-\rho') \|d_1 - d_2\|_{TV}.$$

Finally, we state the approximation bounds for the approximate TD-learning algorithm

**Lemma 4** *Suppose Assumptions 1 holds and consider that $\mu(a \mid B) \ge \delta \, \forall \, a, B$, such that [9] $(1-\rho)^l < \delta |\mathbb{A}|$ for all policies $\mu$. Let $\bar{Q}^\mu_{\tau + l'}$ denote the Q-function obtained by running Algorithm 1 for $\tau > 4(1-\gamma)^2$ iterations with a fixed stepsize $\epsilon_t = 1/\sqrt{\tau}$, and $l' = \frac{\log \tau}{2 \log(1-\rho')}$. Moreover, let $\tilde{Q}^\mu$ be the actual Q-function of the Superstate MDP corresponding to the policy $\mu$, which satisfies $\tilde{Q}^\mu(B,a) = \tilde{T}^\mu(\tilde{Q}^\mu(B,a)) \, \forall B \in \mathbb{H}_{\le l}, a \in \mathcal{A}$ and let $\hat{\theta} := \min_{\|\theta\| \le R} \|\tilde{Q}^\mu - \Phi^T \theta\|^2$. Then,*

$$\mathbb{E}\|\bar{Q}^\mu_{\tau+l'} - \tilde{Q}^\mu\|_\infty \le \|\Phi^T \hat{\theta} - \tilde{Q}^\mu\|_\infty + \left(1 - \frac{2(1-\gamma)}{\sqrt{\tau}}\right)^\tau \|\theta_l - \hat{\theta}\|_2 + \left[\frac{1 - (1 - 2(1-\gamma)/\sqrt{\tau})^\tau}{(1-\gamma)}\right]$$

$$\times \left[\frac{(\bar{r} + 2R)^2}{2\sqrt{\tau}} + \frac{C_2(\bar{r} + 2R)\log \tau}{\sqrt{\tau} \log(1-\rho')} + (1-\rho')^{\frac{\log \tau}{2\log(1-\rho')}}\left(R\bar{r} + R^2(1 + (1-\rho)\gamma)\right)\right.$$

$$\left. + 2R\bar{r}(1-\rho)^l + \frac{2}{\rho'}(1-\rho)^l\left(R\bar{r} + R^2(1 + (1-\rho)\gamma)\right)\right] := \xi_{TD\text{-}Error},$$

---

**Algorithm 2: A Policy Optimization Based Algorithm to learn the Superstate MDP**

Set $Q_0(B,a) = 0, \; \forall B \in \mathbb{H}_{\le l}, a \in \mathcal{A}$
**for** $i = 1, 2, \ldots, M$ **do**
    $\mu_i(a \mid B) \propto \exp\left(\eta \sum_{j=1}^i \bar{Q}^{\mu_{j-1}}_{\tau+l'}(B,a)\right)$
    Initialize $\theta_l$ randomly in $\mathcal{B}(R)$
    Sample $s_0 \sim \mathcal{D}$ and set $H_0^i = \{\}$
    **for** $t = 0$ *to* $\tau + l' - 1$ **do**
        Select action $a_t$ according to policy $\mu_i(\cdot \mid \mathcal{G}(H_t^i))$
        Observe reward $r_t$ and the next observation $y_{t+1}$
        Update the history $H_{t+1}^i = H_t^i \| \{a_t, y_{t+1}\}$
        Select action $a_{t+1}$ according to the policy $\mu_i(\cdot \mid \mathcal{G}(H_{t+1}^i))$
        **if** $t \ge l'$ **then**
            $\theta_{t+1/2} = \theta_t + \epsilon_t\left(r_t + \gamma \phi^T\big(\mathcal{G}(H_{t+1}^i), a_{t+1}\big)\theta_t - \phi^T\big(\mathcal{G}(H_t^i), a_t\big)\theta_t\right)\phi(\mathcal{G}(H_t^i), a_t)$
            $\theta_{t+1} = \text{Proj}_{\mathcal{B}(R)}(\theta_{t+1/2})$
        **end**
    **end**
    $\bar{Q}^{\mu_i}_{\tau+l'}(B,a) = \Phi^T(B,a)\theta_{\tau+l'}$
**end**

---

Next, we combine Algorithm 1 with the POLITEX algorithm from Abbasi-Yadkori et al. [2019] for the policy update rule to learn the optimal policy. Note that the POLITEX algorithm is proposed for the average reward setting, whereas in this work, we extend their analysis to the discounted reward problem. The overall algorithm is outlined in Algorithm 2, where the inner loop performs TD learning, as described in Algorithm 1, while the outer loop performs policy updates using an exponential update rule which incorporates aggregate information from learned Q-functions.

---

[9] Given how the policy is chosen in Algorithm 2, a non-zero $\delta$ always exists

Next, to evaluate the performance of our algorithm, we use the following notion of *regret*. Similar definitions have also been used in He et al. [2021]. Regret is therefore defined as

$$\mathcal{R}_T = \mathbb{E}\Big[\sum_{i=1}^{M}\sum_{j=0}^{\tau+l'-1}\Big(V^{\mu^*}(\boldsymbol{\pi}(H_0))-\tilde{V}^{\mu_i}(\mathcal{G}(H_0))\Big)\Big] = (\tau+l')\sum_{i=1}^{M}\mathbb{E}\Big[V^{\mu^*}(\boldsymbol{\pi}(H_0))-\tilde{V}^{\mu_i}(\mathcal{G}(H_0))\Big],$$
(9)

where $V^{\mu^*}$ and $\tilde{V}^{\mu_i}$ are value functions of the actual POMDP and the Superstate MDP under optimal policy $\mu^*$ and policy $\mu_i$, respectively. Here, the number of policy updates $M$ and the number of inner TD learning iterations in each episode $\tau$ are chosen such that $M(\tau + l') = T$. The intuition behind the definition is that, suppose the algorithm stops at the $j$-th inner iteration of the $i$-th policy update episode. Then, the error between the expected discounted reward corresponding to the optimal policy and the policy output by the algorithm is $V^{\mu^*}(\boldsymbol{\pi}(H_0)) - \tilde{V}^{\mu_i}(\mathcal{G}(H_0))$. Therefore, $\mathcal{R}_T/T$ can also be viewed as the expected error incurred by the algorithm if it stops at a uniformly chosen random time.

The following theorem provides an analytical upper bound on the regret of our proposed algorithm.[10]

**Theorem 3** *Let $V^*$ be the optimal value function of the POMDP, and $\{\mu_i\}_{i=1}^{M}$ be the policies learned in Algorithm 2 at the corresponding discrete time intervals $t_i = [(i-1)(\tau+l')+1, i(\tau+l')]$, $i = 1,\ldots,M$. Moreover, let the regret $\mathcal{R}_T$ be as defined in Eq. (9). Further, let $\tau = \sqrt{T}$ and thus $l' = \frac{\log T}{4\log(1-\rho')}$ and $M = \frac{T}{(\tau+l')}$. Then, the regret is bounded as*

$$\mathcal{R}_T \le T \cdot (\xi_{FA} + \xi_{HA}) + \mathcal{O}(T^{3/4}\log T)$$
(10)

*where*

$$\xi_{FA} = 2\sum_{i=1}^{M}\|\Phi^T\hat{\theta}_i - \tilde{Q}^{\mu_i}\|_\infty/M,$$

$$\xi_{HA} = (1-\rho)^l\Big[\frac{1-(1-2(1-\gamma)/\sqrt{\tau})^\tau}{(1-\gamma)}\Big]\cdot\Big(4R\bar{r} + 4/\rho'\big(R\bar{r} + R^2(1+(1-\rho)\gamma)\big)\Big)$$

$$+ \frac{2\bar{r}}{(1-\gamma)} + \frac{2\bar{r}\gamma}{(1-\gamma)\big(2(1-\gamma)+(1-\rho)^l\gamma\big)}\Big),$$

Since Algorithm 2 is devised to optimize the Superstate MDP, a small regret implies that the realized trajectory of the algorithm under the actual POMDP is also close to its optimal value.

Note that $\xi_{FA}$ is the error due to linear function approximation which can be reduced by using a good set of feature vectors. Similarly, $\xi_{HA}$ is the error due to approximating the history using a truncated history of length $l$ (Superstate), which quantifies the tradeoff between increased complexity in terms of the number of states in the Superstate MDP and the approximation error.

# 5   Conclusion

We show that standard policy optimization algorithms can effectively approximate an optimal POMDP policy by modeling it as an MDP over finite histories, and provide convergence guarantees without the heavy computational cost or restrictive assumptions of prior methods. Our results also extend to the linear function approximation setting, ensuring scalability to large state spaces. Finally, we extend the POLITEX algorithm to the discounted reward setting and analyzed the regret with respect to the optimal POMDP value function. Overall, our work provides tighter theoretical guarantees, improved efficiency, and a more scalable solution for solving PORL problems. Future work could focus on tightening the approximation bounds by leveraging more expressive function approximators, such as LSTMs or Transformer-based architectures.

## Acknowledgments and Disclosure of Funding

The work done in this paper was supported by NSF grants CNS 23-12714 and CCF 22-07547 and AFOSR grants FA9550-24-1-0002 and FA9550-23-1-0107. Additionally, Ameya was also supported by the Henderson Fellowship.

---

[10]A direct comparison of our bound with Cayci et al. [2024] is also provided in the Appendix.

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

# A  Appendix A: Omitted Proofs

## A.1  Proof of Lemma 1

Let $H$ and $H'$ be two histories with the same superstate, $\mathcal{G}(H') = \mathcal{G}(H) = \{a_1, y_1, a_2, \ldots, y_{(l-1)}, a_l, y_l\}$. Moreover, let $t_1$ and $t_2$ be lengths of $H$ and $H'$, respectively. Then, we have

$$\boldsymbol{\pi}(H) = K_{a_l, y_l} \otimes \ldots \otimes K_{a_1, y_1} \otimes \boldsymbol{\pi}(H_{t_1 - l}),$$
$$\boldsymbol{\pi}(H') = K_{a_l, y_l} \otimes \ldots \otimes K_{a_1, y_1} \otimes \boldsymbol{\pi}(H'_{t_2 - l}).$$

Using Assumption 1 inductively, we can write

$$\|\boldsymbol{\pi}(H) - \boldsymbol{\pi}(H')\|_{TV} = \|K_{a_l, y_l} \otimes \ldots \otimes K_{a_1, y_1} \otimes \boldsymbol{\pi}(H_{t_1 - l}) - K_{a_l, y_l} \otimes \ldots \otimes K_{a_1, y_1} \otimes \boldsymbol{\pi}(H'_{t_2 - l})\|_{TV}$$
$$\leq (1 - \rho)^l \|\boldsymbol{\pi}(H_{t_1 - l}) - \boldsymbol{\pi}(H'_{t_2 - l})\|_{TV}$$
$$\leq (1 - \rho)^l.$$

## A.2  Proof of Lemma 2

Without loss of generality, let us assume $\sum_i a_i b_i - \sum_i c_i d_i \geq 0$. We have

$$\sum_i a_i b_i - \sum_i c_i d_i$$
$$= \sum_i \left[ \frac{(a_i - c_i)(b_i + d_i)}{2} + \frac{(b_i - d_i)(a_i + c_i)}{2} \right]$$
$$\leq \sum_i \left[ \frac{(a_i - c_i)(b_i + d_i)}{2} + \frac{\mid b_i - d_i \mid (a_i + c_i)}{2} \right]$$
$$= \sum_{i: a_i \geq c_i} \left[ (a_i - c_i)\left( \max(b_i, d_i) - \frac{\mid b_i - d_i \mid}{2} \right) + \mid b_i - d_i \mid \left( \frac{a_i + c_i}{2} \right) \right]$$
$$+ \sum_{i: c_i > a_i} \left[ (a_i - c_i)\left( \frac{b_i + d_i}{2} \right) + \mid b_i - d_i \mid \left( c_i - \frac{(c_i - a_i)}{2} \right) \right]$$
$$\leq \sum_{i: a_i \geq c_i} \left[ (a_i - c_i)\left( \max(b_i, d_i) - \frac{\mid b_i - d_i \mid}{2} \right) + \mid b_i - d_i \mid \left( \frac{a_i + c_i}{2} \right) \right]$$
$$+ \sum_{i: c_i > a_i} \mid b_i - d_i \mid \left( c_i - \frac{(c_i - a_i)}{2} \right)$$
$$\leq \sum_{i: a_i \geq c_i} \left[ (a_i - c_i) \max(b_i, d_i) + c_i \mid b_i - d_i \mid \right] + \sum_{i: c_i > a_i} \mid b_i - d_i \mid \left( c_i - \frac{(c_i - a_i)}{2} \right)$$
$$\leq \sum_{i: a_i \geq c_i} \left[ (a_i - c_i) \max(\|b\|_\infty, \|d\|_\infty) + c_i \|b - d\|_\infty \right] + \sum_{i: c_i > a_i} \|b - d\|_\infty \left( c_i - \frac{(c_i - a_i)}{2} \right)$$
$$= \sum_{i: a_i \geq c_i} (a_i - c_i) \max(\|b\|_\infty, \|d\|_\infty) - \sum_{i: c_i > a_i} \|b - d\|_\infty \left( \frac{c_i - a_i}{2} \right) + \|b - d\|_\infty$$
$$= \frac{\|a - c\|_1}{2} \max(\|b\|_\infty, \|d\|_\infty) - \|b - d\|_\infty \frac{\|a - c\|_1}{4} + \|b - d\|_\infty,$$

where the last equality follows from relations $\sum_{i: a_i \geq c_i} (a_i - c_i) + \sum_{i: c_i > a_i} (c_i - a_i) = \|a - c\|_1$ and $\sum_{i: a_i \geq c_i} (a_i - c_i) - \sum_{i: c_i > a_i} (c_i - a_i) = \sum_i a_i - \sum_i c_i = 0$, which together implies that $\sum_{i: a_i \geq c_i} (a_i - c_i) = \sum_{i: c_i > a_i} (c_i - a_i) = \frac{\|a - c\|_1}{2}$.

### A.3 Proof of Theorem 2

Consider an arbitrary history $H \in \mathbb{H}$ with the corresponding superstate $\mathcal{G}(H) = B$. For any $a \in \mathcal{A}$,

$$
\left| \tilde{r}(B, a) - \sum_s \pi(s \mid H) r(s, a) \right| = \left| \sum_s \pi(s \mid B) r(s, a) - \sum_s \pi(s \mid H) r(s, a) \right|
$$
$$
\leq \sum_s \left| \pi(s \mid B) - \pi(s \mid H) \right| |r(s, a)|
$$
$$
\leq 2(1 - \rho)^l \bar{r}, \tag{11}
$$

where the last inequality follows from Lemma 1.

Let $\delta := \|V^*\big(\boldsymbol{\pi}(H)\big) - \tilde{V}\big(\mathcal{G}(H)\big)\|_\infty$, and note that $\delta$ is finite since the value functions are finite.

$$
\delta = \|V^*\big(\boldsymbol{\pi}(H)\big) - \tilde{V}\big(\mathcal{G}(H)\big)\|_\infty = \|V^*\big(\boldsymbol{\pi}(H)\big) - \tilde{V}(B)\|_\infty \tag{12}
$$

Next, we bound $|V^*\big(\boldsymbol{\pi}(H)\big) - \tilde{V}\big(\mathcal{G}(H)\big)|$ for all $H$. Using Bellman's Optimality equation for the belief state MDP (POMDP) and superstate MDP, we have

$$
|V^*\big(\boldsymbol{\pi}(H)\big) - \tilde{V}\big(\mathcal{G}(H)\big)|
$$
$$
= \max_a \left\{ \sum_s \pi(s \mid H) r(s, a) + \gamma \sum_y V^*\big(\boldsymbol{\pi}(H \| \{y, a\})\big) \sigma(\boldsymbol{\pi}(H), y, a) \right\}
$$
$$
- \max_a \left\{ \tilde{r}(B, a) + \gamma \sum_y \tilde{V}\big(\mathcal{G}(B \| \{y, a\})\big) \sigma(\boldsymbol{\pi}(B), y, a) \right\}.
$$

Suppose $\hat{a}$ is the best action for the POMDP at belief state $\boldsymbol{\pi}(H)$. Then we can write $|V^*\big(\boldsymbol{\pi}(H)\big) - \tilde{V}\big(\mathcal{G}(H)\big)|$ as

$$
|V^*\big(\boldsymbol{\pi}(H)\big) - \tilde{V}\big(\mathcal{G}(H)\big)|
$$
$$
= \sum_s \pi(s \mid H) r(s, \hat{a}) + \gamma \sum_y V^*\big(\boldsymbol{\pi}(H \| \{y, \hat{a}\})\big) \sigma(\boldsymbol{\pi}(H), y, \hat{a})
$$
$$
- \max_a \left\{ \tilde{r}(B, a) + \gamma \sum_y \tilde{V}\big(\mathcal{G}(B \| \{y, a\})\big) \sigma(\boldsymbol{\pi}(B), y, a) \right\}. \tag{13}
$$

Now, since the maximum value of the second term will be greater than evaluating the second term for $a = \hat{a}$, we have

$$
|V^*\big(\boldsymbol{\pi}(H)\big) - \tilde{V}\big(\mathcal{G}(H)\big)| \leq \left( \sum_s \pi(s \mid H) r(s, \hat{a}) - \tilde{r}(B, \hat{a}) \right)
$$
$$
+ \gamma \left[ \sum_y V^*\big(\boldsymbol{\pi}(H \| \{y, \hat{a}\})\big) \sigma(\boldsymbol{\pi}(H), y, \hat{a}) - \sum_y \tilde{V}\big(\mathcal{G}(B \| \{y, \hat{a}\})\big) \sigma(\boldsymbol{\pi}(B), y, \hat{a}) \right]
$$
$$
\leq 2(1 - \rho)^l \bar{r} + \gamma \left[ \sum_y V^*\big(\boldsymbol{\pi}(H \| \{y, \hat{a}\})\big) \sigma(\boldsymbol{\pi}(H), y, \hat{a}) - \sum_y \tilde{V}\big(\mathcal{G}(B \| \{y, \hat{a}\})\big) \sigma(\boldsymbol{\pi}(B), y, \hat{a}) \right],
$$

where in the second inequality we have used (11).

Next, we note that $\sum_y \sigma(\boldsymbol{\pi}(H), y, \hat{a}) = \sum_y \sigma(\boldsymbol{\pi}(B), y, \hat{a}) = 1$, as $\sigma(\cdot)$ is a probability distribution over the observation set $\mathcal{Y}$. Moreover, by the definition of $\delta$ and since $\max\big(\|\tilde{V}\|_\infty, \|V^*\|_\infty\big) \leq \frac{\bar{r}}{1 - \gamma}$, we can use Lemma 2 to upper-bound the second term in the above expression and obtain

$$
|V^*\big(\boldsymbol{\pi}(H)\big) - \tilde{V}\big(\mathcal{G}(H)\big)| \leq 2(1 - \rho)^l \bar{r}
$$
$$
+ \gamma \left[ \|\sigma(\boldsymbol{\pi}(H), \cdot, \hat{a}) - \sigma(\boldsymbol{\pi}(B), \cdot, \hat{a})\|_{TV} \frac{\bar{r}}{1 - \gamma} + \delta - \frac{\delta}{2} \|\sigma(\boldsymbol{\pi}(H), \cdot, \hat{a}) - \sigma(\boldsymbol{\pi}(B), \cdot, \hat{a})\|_{TV} \right]
$$
$$
= 2(1 - \rho)^l \bar{r} + \gamma \left[ \|\sigma(\boldsymbol{\pi}(H), \cdot, \hat{a}) - \sigma(\boldsymbol{\pi}(B), \cdot, \hat{a})\|_{TV} \left( \frac{\bar{r}}{1 - \gamma} - \frac{\delta}{2} \right) + \delta \right]. \tag{14}
$$

Note that the above inequality holds for all $H$.

Furthermore, we can bound the total variation norm in the above relation as

$$
\begin{aligned}
\left\|\sigma(\boldsymbol{\pi}(H), \cdot, \hat{a}) - \sigma(\boldsymbol{\pi}(B), \cdot, \hat{a})\right\|_{TV} &= \sum_y \left|\sigma(\hat{a}, y, \boldsymbol{\pi}(H)) - \sigma(\hat{a}, y, \boldsymbol{\pi}(B))\right| \\
&= \sum_y \left|\sum_{s,s'} \Phi(y \mid s')\mathcal{P}(s' \mid s, \hat{a})(\pi(s \mid H) - \pi(s \mid B))\right| \\
&\leq \sum_s \left|\pi(s \mid H) - \pi(s \mid B)\right| \sum_{y,s'} \Phi(y \mid s')\mathcal{P}(s' \mid s, \hat{a}) \\
&= 2\|\boldsymbol{\pi}(H) - \boldsymbol{\pi}(B)\|_{TV} \leq 2(1 - \rho)^l,
\end{aligned}
$$

where the first inequality holds by the triangle inequality, and the second inequality is obtained using Lemma 1. Now, we consider two cases. Case 1: $\frac{\bar{r}}{1-\gamma} - \frac{\delta}{2} \geq 0$

Therefore, we have

$$
\delta \leq 2(1-\rho)^l \bar{r} + \gamma\left[\sup_H \left\|\sigma(\boldsymbol{\pi}(H), \cdot, \hat{a}) - \sigma(\boldsymbol{\pi}(B), \cdot, \hat{a})\right\|_{TV}\left(\frac{\bar{r}}{1-\gamma} - \frac{\delta}{2}\right) + \delta\right] \tag{15}
$$

Case 2: $\frac{\bar{r}}{1-\gamma} - \frac{\delta}{2} < 0$

Therefore, we have

$$
\delta \leq 2(1-\rho)^l \bar{r} + \gamma\left[\inf_H \left\|\sigma(\boldsymbol{\pi}(H), \cdot, \hat{a}) - \sigma(\boldsymbol{\pi}(B), \cdot, \hat{a})\right\|_{TV}\left(\frac{\bar{r}}{1-\gamma} - \frac{\delta}{2}\right) + \delta\right] \tag{16}
$$

Now, both $\inf_H \left\|\sigma(\boldsymbol{\pi}(H), \cdot, \hat{a}) - \sigma(\boldsymbol{\pi}(B), \cdot, \hat{a})\right\|_{TV}$ and $\sup_H \left\|\sigma(\boldsymbol{\pi}(H), \cdot, \hat{a}) - \sigma(\boldsymbol{\pi}(B), \cdot, \hat{a})\right\|_{TV}$ are less than $2(1-\rho)^l$.

Therefore, we can write

$$
\begin{aligned}
\delta &\leq 2(1-\rho)^l \bar{r} + \gamma\left[2(1-\rho)^l\left(\frac{\bar{r}}{1-\gamma} - \frac{\delta}{2}\right) + \delta\right] \\
&\leq \frac{2(1-\rho)^l \bar{r}}{1-\gamma} + \frac{\gamma}{1-\gamma}\left[2(1-\rho)^l\left(\frac{\bar{r}}{1-\gamma} - \frac{\delta}{2}\right)\right] \\
\delta\left(1 + \frac{\gamma(1-\rho)^l}{1-\gamma}\right) &\leq \frac{2(1-\rho)^l \bar{r}}{1-\gamma} + \frac{2\gamma(1-\rho)^l \bar{r}}{(1-\gamma)^2} \\
\delta &\leq \frac{2(1-\rho)^l \bar{r}}{(1-\gamma)(1 + \frac{\gamma(1-\rho)^l}{1-\gamma})} + \frac{2\gamma\bar{r}(1-\rho)^l}{(1-\gamma)(1 - \gamma + \gamma(1-\rho)^l)} \\
&\leq \frac{2(1-\rho)^l \bar{r}}{(1-\gamma)} + \frac{2\gamma\bar{r}(1-\rho)^l}{(1-\gamma)(1 - \gamma + \gamma(1-\rho)^l)} \tag{17}
\end{aligned}
$$

## A.4   Corollary 1

Additionally, if $N$ is the number of states of the Superstate MDP. Then, we can state the following result:

**Corollary 1** *If $N$ is the number of states in the Superstate MDP, then the difference of the optimal value functions in Theorem 2 can be upper-bounded as*

$$
\left\|V^{\mu^*}(\boldsymbol{\pi}(H)) - \tilde{V}(\mathcal{G}(H))\right\|_\infty \leq \frac{2\bar{r}(1-\rho)^{-1}N^{-\kappa}}{1-\gamma} + \frac{4\bar{r}\gamma(1-\rho)^{-1}N^{-\kappa}}{(1-\gamma)(2(1-\gamma) + \gamma N^{-\kappa})},
$$

*where* $\kappa = \frac{\log(1/(1-\rho))}{\log(|\mathcal{Y}||\mathcal{A}|)}$.

**Proof:**
For a fixed $l$, the number of superstates can be all possible histories of length at most $l$. Thus,

$$
N = 1 + |\mathcal{Y}||\mathcal{A}| + |\mathcal{Y}|^2|\mathcal{A}|^2 + \ldots + |\mathcal{Y}|^l|\mathcal{A}|^l < |\mathcal{Y}|^{l+1}|\mathcal{A}|^{l+1},
$$

which implies $l + 1 > \frac{\log N}{\log(|\mathcal{Y}||\mathcal{A}|)}$. Therefore, $(1-\rho)^l < (1-\rho)^{-1} \cdot N^{\frac{\log(1-\rho)}{\log(|\mathcal{Y}||\mathcal{A}|)}}$. Similarly, since $l < \frac{\log N}{\log(|\mathcal{Y}||\mathcal{A}|)}$, we have $(1-\rho)^l > N^{\frac{\log(1-\rho)}{\log(|\mathcal{Y}||\mathcal{A}|)}}$. Substituting these relations into Eq. (8) gives us the desired result.

## A.5 Proof of Lemma 3

Let $d_1, d_2$ be distributions over the superstates and similarly $e_1, e_2$ be the canonical basis vectors in $\mathbb{R}^{|\mathbb{H}_{\leq l}|}$. Then there exists $\alpha_{i,j}$ such that $\alpha_{i,j} \geq 0$ and $\sum_{i,j} \alpha_{i,j} = \|d_1 - d_2\|_{TV}$ and the following holds

$$
\begin{aligned}
\left\|\tilde{\mathcal{P}}^\mu d_1 - \tilde{\mathcal{P}}^\mu d_2\right\|_{TV} &= \|\tilde{\mathcal{P}}^\mu(d_1 - d_2)\|_{TV} \\
&= \|\sum_{i,j} \alpha_{i,j} \tilde{\mathcal{P}}^\mu(e_i - e_j)\|_{TV} \\
&\leq \sum_{i,j} \alpha_{i,j} \|\tilde{\mathcal{P}}^\mu(e_i - e_j)\|_{TV}
\end{aligned}
\tag{18}
$$

Note that the maximum value of the above term is $\sum_{i,j}$ which is $\|d_1 - d_2\|_{TV}$. Therefore, if for some $i,j$ we have $\alpha_{i,j} > 0$ and $\|\tilde{\mathcal{P}}^\mu(e_i - e_j)\|_{TV} < 1$, we are guaranteed a contraction.

Now suppose that $e_i$ corresponds to a superstate $B_i$ and $e_j$ corresponds to a superstate $B_j$

$$
\begin{aligned}
&\|\tilde{\mathcal{P}}^\mu(e_i - e_j)\|_{TV} \\
=&\| \sum_a \mu(a \mid B_i) \sum_{y,s,s'} \mathbb{I}[\mathcal{G}(B_i \,\|\, \{y,a\}) = B'] \cdot \Phi(y \mid s')\mathcal{P}(s' \mid s,a)\pi(s \mid B_i) \\
&- \sum_a \mu(a \mid B_j) \sum_{y,s,s'} \mathbb{I}[\mathcal{G}(B_j \,\|\, \{y,a\}) = B'] \cdot \Phi(y \mid s')\mathcal{P}(s' \mid s,a)\pi(s \mid B_j)\|_{TV}
\end{aligned}
\tag{19}
$$

Next, we will show that if $B_i$ and $B_j$ are two superstates which differ in the first two elements then $\|\tilde{\mathcal{P}}^\mu(B_i - B_j)\|_{TV} < 1$.

Therefore, we focus on pairs of $B_i, B_j$ such that $\mathcal{G}(B_i \,\|\, \{y,a\}) = \mathcal{G}(B_j \,\|\, \{y,a\})$, i.e., $B_i, B_j$ which only differ in the first two elements. For such a pair, we can simplify further

$$
\begin{aligned}
&\| \sum_a \mu(a \mid B_i) \sum_{y,s,s'} \mathbb{I}[\mathcal{G}(B_i \,\|\, \{y,a\}) = B'] \cdot \Phi(y \mid s')\mathcal{P}(s' \mid s,a)\pi(s \mid B_i) \\
&- \sum_a \mu(a \mid B_j) \sum_{y,s,s'} \mathbb{I}[\mathcal{G}(B_j \,\|\, \{y,a\}) = B'] \cdot \Phi(y \mid s')\mathcal{P}(s' \mid s,a)\pi(s \mid B_j)\|_{TV} \\
=& \|\{\mu(a \mid B_i) \sum_{s,s'} \Phi(y \mid s')\mathcal{P}(s' \mid s,a)\pi(s \mid B_i) - \mu(a \mid B_j) \sum_{s,s'} \Phi(y \mid s')\mathcal{P}(s' \mid s,a)\pi(s \mid B_j)\}_{y,a}\|_{TV} \\
=& 1/2 \sum_{y,a} |\mu(a \mid B_i) \sum_{s,s'} \Phi(y \mid s')\mathcal{P}(s' \mid s,a)\pi(s \mid B_i) - \mu(a \mid B_j) \sum_{s,s'} \Phi(y \mid s')\mathcal{P}(s' \mid s,a)\pi(s \mid B_j)| \\
\leq& 1/2 \sum_{y,a} \left[\left|\mu(a \mid B_j)\Big[\sum_{s,s'} \Phi(y \mid s')\mathcal{P}(s' \mid s,a)\pi(s \mid B_i) - \sum_{s,s'} \Phi(y \mid s')\mathcal{P}(s' \mid s,a)\pi(s \mid B_j)\Big]\right| \right. \\
&+ \left. \left|(\mu(a \mid B_i) - \mu(a \mid B_j)) \sum_{s,s'} \Phi(y \mid s')\mathcal{P}(s' \mid s,a)\pi(s \mid B_i)\right|\right] \\
\leq& \|\pi(s \mid B_i) - \pi(s \mid B_j)\|_{TV} + \|\mu(a \mid B_i) - \mu(a \mid B_j)\|_{TV} \\
\leq& (1-\rho)^l + 1 - |\mathbb{A}|\delta
\end{aligned}
\tag{20}
$$

For the last inequality we assume that our policy has a small exploration component such that $\mu(a \mid B) \geq \delta \,\forall\, B$. Therefore, for sufficiently large horizon length $l$ or exploration $\delta$, assumption 2 is automatically satisfied.

Therefore if $\alpha_{i,j} > 0$ for pairs of $B_i, B_j$ such that $\mathcal{G}(B_i \,\|\, \{y,a\}) = \mathcal{G}(B_j \,\|\, \{y,a\})$ we are guaranteed a contraction

Next, we will show that we can construct an algorithm such that $\alpha_{i,j} > 0$ for all such pairs $B_i, B_j$.

To construct $\alpha_{i,j}$ the following greedy algorithm can be used. Let $v_1, v_2 \in \mathbb{R}^n$ be two probability distributions. Define the difference vector $d = v_1 - v_2$, and define:

---

**Algorithm 3: A Greedy Algorithm to construct $\alpha$**

**Input:** Two discrete distributions $v_1 = (v_1(1), v_1(2), \ldots, v_1(n))$ and $v_2 = (v_2(1), v_2(2), \ldots, v_2(n))$
Compute the difference vector $\Delta = v_1 - v_2$ where $\Delta_i = v_1(i) - v_2(i)$
Define the surplus set $S = \{i \mid \Delta_i > 0\}$ and the deficit set $D = \{j \mid \Delta_j < 0\}$
Initialize $\alpha_{ij} = 0$ for all $i, j$
**for** *each* $i \in S$ **do**

**end**
each $j \in D$ $\alpha_{ij} \leftarrow \min(\Delta_i, -\Delta_j)$
$\Delta_i \leftarrow \Delta_i - \alpha_{ij}$
$\Delta_j \leftarrow \Delta_j + \alpha_{ij}$

---

See that first $\alpha_{ij} = \min(\Delta_i, -\Delta_j)$. Therefore, since $\Delta_i, \Delta_j \neq 0$, $\alpha_{ij} > 0$. Additionally, the steps $\Delta_i \leftarrow \Delta_i - \alpha_{ij}$, $\Delta_j \leftarrow \Delta_j + \alpha_{ij}$ ensures that either one of them goes to 0 and is removed from the surplus/deficit set.

Additionally, it is straightforward to see that $\sum_i = \|d\|$. Thus, $\Delta_i, \Delta_j \neq 0 \implies \alpha_{ij} > 0$

## A.6 Proof of Lemma 4

We first introduce some notations that will be used to prove the result. Let us define

$$g_t(\theta) := \left[ r_t + \gamma \phi^T\big(\mathcal{G}(H_{t+1}), a_{t+1}\big)\theta - \phi^T\big(\mathcal{G}(H_t), a_t\big)\theta \right] \cdot \phi(\mathcal{G}(H_t), a_t),$$

and note that it can be written in a compact form as $g_t(\theta) = \Phi R_t + \gamma \Phi E_t \Phi^T \theta - \Phi D_t \Phi^T \theta$, where[11]

$$D_t := \mathrm{diag}\left( \big[\mathbb{I}[(B_t, a_t) = (B, a)]\big]_{B \in \mathbb{H}_{\leq l}, a \in \mathcal{A}} \right),$$
$$R_t := \big[ r_t \mathbb{I}[(B_t, a_t) = (B, a)] \big]^T_{B \in \mathbb{H}_{\leq l}, a \in \mathcal{A}},$$
$$E_t\big((B, a), (B', a')\big) := \mathbb{I}[(B_t, a_t) = (B, a)] \cdot \mathbb{I}[(B_{t+1}, a_{t+1}) = (B', a')],$$
$$\Phi = [\phi(B, a)]_{B \in \mathbb{H}_{\leq l}, a \in \mathcal{A}}.$$

Additionally, let $\bar{g}(\theta) := \Phi \tilde{D}^\mu \tilde{r} + \gamma \Phi \tilde{D}^\mu \tilde{P}^\mu \Phi^T \theta - \Phi \tilde{D}^\mu \Phi^T \theta$, where $\tilde{D}^\mu$ and $\tilde{P}^\mu$ denote the stationary distribution and the state transition matrix for the Superstate MDP when following policy $\mu$, respectively. In particular, $\tilde{P}^\mu(B', a' \mid B, a) = \mu(a' \mid B)\tilde{\mathcal{P}}(B' \mid B, a)$, where $\tilde{\mathcal{P}}(B' \mid B, a)$ is given by (5). Similarly $\tilde{r} := [r(B, a)]^T_{B \in \mathbb{H}_{\leq l}, a \in \mathcal{A}}$ is the vector of rewards corresponding to the Superstate MDP as defined in (4). Finally, we also define

$$\eta_t(\theta) = (\theta - \hat{\theta})^T (g_t(\theta) - \bar{g}(\theta)).$$

Next, we state and prove an auxiliary lemma that would be required for our main analysis.

**Lemma 5** *The following inequalities are true:*[12]

  (a) $|\eta_t(\theta_t)| \leq C_1$, *where* $C_1 = 2R \cdot 2(\bar{r} + 2R)$.

  (b) $\|g_t(\theta_1) - g_t(\theta_2)\| \leq C_2 \|\theta_1 - \theta_2\| \; \forall \theta_1, \theta_2$, *where* $C_2 = (2\bar{r} + 12R)$.

---

[11] $[v]_{v \in V}$ denotes a matrix obtained by concatenating all vectors in $V$, where each $v$ is a column vector of the matrix.

[12] Unless stated, the norms are $L_2$-norms.

**Proof:**
To show part (a), using the Cauchy-Schwarz inequality, we have

$$|\eta_t(\theta_t)| \leq \|(\theta_t - \hat{\theta})\|\|(g_t(\theta_t) - \bar{g}(\theta_t))\| \leq 2R \cdot 2(\bar{r} + 2R).$$

To prove part (b), for simplicity let $B = \mathcal{G}(H_t), a_t = a$, and $B' = \mathcal{G}(H_{t+1}), a_{t+1} = a'$. Then,

$$\|g_t(\theta_1) - g_t(\theta_2)\| = \|(\gamma\phi^T(B', a') - \phi^T(B, a))(\theta_1 - \theta_2)\phi(B, a)\|$$
$$\leq \|\theta_1 - \theta_2\|\|\phi(B, a)\|\|\gamma\phi^T(B', a') - \phi^T(B, a)\| \leq 2\|\theta_1 - \theta_2\|$$

Similarly, we can show that $\|\bar{g}(\theta_1) - \bar{g}(\theta_2)\| \leq 2\|\theta_1 - \theta_2\|$. Thus,

$$|\eta_t(\theta_1) - \eta_t(\theta_2)| = |(g_t(\theta_1) - \bar{g}(\theta_1))^T(\theta_1 - \theta_2 + \theta_2 - \hat{\theta}) - ((g_t(\theta_2) - \bar{g}(\theta_2)))^T(\theta_2 - \hat{\theta})|$$
$$\leq \|g_t(\theta_1) - \bar{g}(\theta_1)\|\|\theta_1 - \theta_2\| + \|\theta_2 - \hat{\theta}\|(\|g_t(\theta_1) - g(\theta_2)\| + \|\bar{g}(\theta_1) - \bar{g}(\theta_2)\|)$$
$$\leq (2\bar{r} + 12R)\|\theta_1 - \theta_2\|.$$

We are now ready to prove Lemma 4. Let us consider the Lyapunov function

$$\mathcal{L}(\theta) := \|\theta - \hat{\theta}\|^2.$$

In order to show that $\theta_t$ converges to $\hat{\theta}$, we will show that $\mathcal{L}(\theta_t)$ converges to 0, and obtain finite time bounds for the convergence. To that end, we first relate the successive iterates $\mathcal{L}(\theta_t)$ and $\mathcal{L}(\theta_{t+1})$:

$$\mathcal{L}(\theta_{t+1}) = \|\theta_{t+1} - \hat{\theta}\|^2$$
$$= \|\text{Proj}(\theta_{t+1/2}) - \text{Proj}(\hat{\theta})\|^2$$
$$\leq \|\theta_{t+1/2} - \hat{\theta}\|^2$$
$$= \|\theta_t + \epsilon_t g_t(\theta_t) - \hat{\theta}\|^2$$
$$= \|\theta_t - \hat{\theta}\|^2 + \epsilon_t^2\|g_t(\theta_t)\|^2 + 2\epsilon_t g_t^T(\theta_t)(\theta_t - \hat{\theta})$$
$$\leq \epsilon_t^2(\bar{r} + 2R)^2 + \mathcal{L}(\theta_t) + 2\epsilon_t g_t^T(\theta_t)(\theta_t - \hat{\theta}), \tag{21}$$

where the last step follows from $\|g_t(\theta_t)\| \leq \bar{r} + 2R$. By adding and subtracting $\bar{g}(\theta_t)$ in Eq. (21), we get

$$\mathcal{L}(\theta_{t+1}) \leq \mathcal{L}(\theta_t) + \epsilon_t^2(\bar{r} + 2R)^2 + 2\epsilon_t(\theta_t - \hat{\theta})^T\bar{g}(\theta_t) + 2\epsilon_t(\theta_t - \hat{\theta})^T(g_t(\theta_t) - \bar{g}(\theta_t)).$$

Next, we proceed to bound each of the terms in the above expression. We start by bounding $2\epsilon_t(\theta_t - \hat{\theta})^T\bar{g}_t(\theta_t)$ in terms of $\mathcal{L}(\theta_t)$. We can write

$$2\epsilon_t(\theta_t - \hat{\theta})^T\bar{g}_t(\theta_t)$$
$$= 2\epsilon_t(\theta_t - \hat{\theta})^T\Phi(\tilde{D}^\mu\tilde{r} + \gamma\tilde{D}^\mu\tilde{P}^\mu\Phi^T\theta_t - \tilde{D}^\mu\Phi^T\theta_t)$$
$$= 2\epsilon_t(\Phi^T(\theta_t - \hat{\theta}))^T\tilde{D}^\mu(\tilde{T}^\mu(\Phi^T\theta_t) - \tilde{T}^\mu(\Phi^T\hat{\theta})) + 2\epsilon_t(\Phi^T(\theta_t - \hat{\theta}))^T\tilde{D}^\mu(\Phi^T\hat{\theta} - \Phi^T\theta_t)$$
$$\leq 2\epsilon_t\|\Phi^T(\theta_t - \hat{\theta})\|_{\tilde{D}^\mu}\|\tilde{T}^\mu(\Phi^T\theta_t) - \tilde{T}^\mu(\Phi^T\hat{\theta})\|_{\tilde{D}^\mu} - 2\epsilon_t\|\Phi^T(\theta_t - \hat{\theta})\|_{\tilde{D}^\mu}^2$$
$$\leq 2\epsilon_t(\gamma - 1)\|\Phi^T(\theta_t - \hat{\theta})\|_{\tilde{D}^\mu}^2,$$

where the first inequality uses the Cauchy-Schwarz inequality and the second inequality follows from the contraction property.

Next, we will proceed to bound $\mathbb{E}[2\epsilon_t(\theta_t - \hat{\theta})^T(g_t(\theta_t) - \bar{g}(\theta_t))]$. To that end, we will first relate $\eta_t(\theta_t)$ with $\eta_t(\theta_{t-l'})$. Note that since

$$\|\theta_{t+1} - \theta_t\| = \|\text{Proj}(\theta_t - \epsilon_t g_t(\theta_t)) - \text{Proj}(\theta_t)\|$$
$$\leq \|\epsilon_t g_t(\theta_t)\| = (\bar{r} + 2R)\epsilon_t,$$

we have $\|\theta_t - \theta_{t-l'}\| \leq (\bar{r} + 2R)\sum_{i=t-l'}^{t-1}\epsilon_i$. Therefore, using Lemma 5 (part b), we obtain

$$\eta_t(\theta_t) \leq \eta_t(\theta_{t-l'}) + C_2(\bar{r} + 2R)\sum_{i=t-l'}^{t-1}\epsilon_i.$$

Let $\mathcal{F}_{t-l'} = \{y_0, a_0, r_0, y_1, \ldots, y_t, a_t, r_t, y_{t+1}, a_{t+1}\}$ be the filtration up to time $t - l'$, such that conditioned on $\mathcal{F}_{t-l'}$, $\theta_{t-l'}$ is measurable and deterministic. We can now obtain an upper bound on $\mathbb{E}[\eta_t(\theta_{t-l'})]$ as follows:

$$
\begin{aligned}
\mathbb{E}[\eta_t(\theta_{t-l'})] &= \mathbb{E}\big[\mathbb{E}[\eta_t(\theta_{t-l'}) \mid \mathcal{F}_{t-l'}]\big] \\
&= \mathbb{E}\big[\mathbb{E}\big[\big(\Phi^T(\theta_{t-l'} - \hat{\theta})\big)^T (R_t - \tilde{D}^\mu \tilde{r}) \mid \mathcal{F}_{t-l'}\big]\big] \\
&\quad + \mathbb{E}\big[\mathbb{E}\big[\big(\Phi^T(\theta_{t-l'} - \hat{\theta})\big)^T (\gamma E_t \Phi^T \theta_{t-l'} - \gamma \tilde{D}^\mu \tilde{P}^\mu \Phi^T \theta_{t-l'}) \mid \mathcal{F}_{t-l'}\big]\big] \\
&\quad + \mathbb{E}\big[\mathbb{E}\big[\big(\Phi^T(\theta_{t-l'} - \hat{\theta})\big)^T (\tilde{D}^\mu \Phi^T \theta_{t-l'} - D_t \Phi^T \theta_{t-l'}) \mid \mathcal{F}_{t-l'}\big]\big] \\
&= \mathbb{E}\big[\mathbb{E}\big[\big(\Phi^T(\theta_{t-l'} - \hat{\theta})\big)^T (R_t - D_t \bar{r}) + \big(\Phi^T(\theta_{t-l'} - \hat{\theta})\big)^T (D_t \bar{r} - \tilde{D}^\mu \tilde{r}) \mid \mathcal{F}_{t-l'}\big]\big] \\
&\quad + \mathbb{E}\big[\mathbb{E}\big[\big(\Phi^T(\theta_{t-l'} - \hat{\theta})\big)^T (\gamma E_t \Phi^T \theta_{t-l'} - \gamma \tilde{D}^\mu \tilde{P}^\mu \Phi^T \theta_{t-l'}) \mid \mathcal{F}_{t-l'}\big]\big] \\
&\quad + \mathbb{E}\big[\mathbb{E}\big[\big(\Phi^T(\theta_{t-l'} - \hat{\theta})\big)^T (\tilde{D}^\mu \Phi^T \theta_{t-l'} - D_t \Phi^T \theta_{t-l'}) \mid \mathcal{F}_{t-l'}\big]\big] \\
&\leq 2R\bar{r}(1-\rho)^l + \mathbb{E}\big[\mathbb{E}\big[\big(\Phi^T(\theta_{t-l'} - \hat{\theta})\big)^T (D_t \bar{r} - \tilde{D}^\mu \tilde{r}) \mid \mathcal{F}_{t-l'}\big]\big] \\
&\quad + \mathbb{E}\big[\mathbb{E}\big[\big(\Phi^T(\theta_{t-l'} - \hat{\theta})\big)^T (\gamma E_t \Phi^T \theta_{t-l'} - \gamma \tilde{D}^\mu \tilde{P}^\mu \Phi^T \theta_{t-l'}) \mid \mathcal{F}_{t-l'}\big]\big] \\
&\quad + \mathbb{E}\big[\mathbb{E}\big[\big(\Phi^T(\theta_{t-l'} - \hat{\theta})\big)^T (\tilde{D}^\mu \Phi^T \theta_{t-l'} - D_t \Phi^T \theta_{t-l'}) \mid \mathcal{F}_{t-l'}\big]\big], \quad (22)
\end{aligned}
$$

where the inequality in (22) holds because using a similar argument as in (11), for all $\mathcal{G}(H_t) = B$ and $a_t = a$, we have $r_t - D_t \tilde{r} = \sum_s r(s,a)\pi(s \mid H_t) - \pi(s \mid B) \leq 2\bar{r}(1-\rho)^l$.

Next, let $P_t^\mu$ denote the true probability transition matrix of the POMDP, i.e.,

$$
P_t^\mu(B', a' \mid a, \mathcal{G}(H_t) = B) = \mu(a' \mid B) \sum_{y,s,s'} \mathbb{I}[\mathcal{G}(B \parallel \{y,a\}) = B']\Phi(y \mid s')\mathcal{P}(s' \mid s,a)\pi(s \mid H_t).
$$

We can bound the first term $\mathbb{E}\big[\mathbb{E}\big[\big(\Phi^T(\theta_{t-l'} - \hat{\theta})\big)^T (D_t \bar{r} - \tilde{D}^\mu \tilde{r}) \mid \mathcal{F}_{t-l'}\big]\big]$ in (22) as follows:

$$
\begin{aligned}
&\mathbb{E}\big[\mathbb{E}\big[\big(\Phi^T(\theta_{t-l'} - \hat{\theta})\big)^T (D_t \tilde{r} - \tilde{D}^\mu \tilde{r}) \mid \mathcal{F}_{t-l'}\big]\big] \\
&= \mathbb{E}\big[\big(\Phi^T(\theta_{t-l'} - \hat{\theta})\big)^T \big(\mathbb{E}[D_t \tilde{r} - \tilde{P}^\mu \tilde{D}^\mu \tilde{r} \mid \mathcal{F}_{t-l'}]\big)\big] \\
&= \mathbb{E}\big[\big(\Phi^T(\theta_{t-l'} - \hat{\theta})\big)^T \big(\mathbb{E}[(D_t \tilde{r} - \tilde{P}^\mu D_{t-1} \tilde{r}) \mid \mathcal{F}_{t-l'}] + \mathbb{E}[(\tilde{P}^\mu D_{t-1} \tilde{r} - \tilde{P}^\mu \tilde{D}^\mu \tilde{r}) \mid \mathcal{F}_{t-l'}]\big)\big] \\
&= \mathbb{E}\big[\big(\Phi^T(\theta_{t-l'} - \hat{\theta})\big)^T \big(\mathbb{E}[\mathbb{E}[(D_t \tilde{r} - \tilde{P}^\mu D_{t-1} \tilde{r}) \mid \mathcal{F}_{t-1}] \mid \mathcal{F}_{t-l'}] + \mathbb{E}[(\tilde{P}^\mu D_{t-1} \tilde{r} - \tilde{P}^\mu \tilde{D}^\mu \tilde{r}) \mid \mathcal{F}_{t-l'}]\big)\big] \\
&\leq \mathbb{E}\big[\|\Phi^T(\theta_{t-l'} - \hat{\theta})\|_\infty \|\mathbb{E}[(P_t^\mu D_{t-1} \tilde{r} - \tilde{P}^\mu D_{t-1} \tilde{r}) \mid \mathcal{F}_{t-l'}] + \mathbb{E}[(\tilde{P}^\mu D_{t-1} \tilde{r} - \tilde{P}^\mu \tilde{D}^\mu \tilde{r}) \mid \mathcal{F}_{t-l'}]\|_1\big] \\
&\leq 2R\bar{r}(1-\rho)^l + \mathbb{E}[(\tilde{P}^\mu D_{t-1} \tilde{r} - \tilde{P}^\mu \tilde{D}^\mu \tilde{r}) \mid \mathcal{F}_{t-l'}]\|_1] \\
&\leq 2R\bar{r}(1-\rho)^l + R(1-\rho')\|\mathbb{E}[D_{t-1} \tilde{r} - \tilde{D}^\mu \tilde{r} \mid \mathcal{F}_{t-l'}]\|_1, \quad (23)
\end{aligned}
$$

where the first inequality is derived using the Holder's inequality, and the second inequality holds because

$$
\begin{aligned}
&\|\mathbb{E}\big[(P_t^\mu D_{t-1} \tilde{r} - \tilde{P}^\mu D_{t-1} \tilde{r}) \mid \mathcal{F}_{t-l'}\big]\|_1 \\
&\leq \mathbb{E}\big[\|(P_t^\mu D_{t-1} \tilde{r} - \tilde{P}^\mu D_{t-1} \tilde{r})\|_1 \mid \mathcal{F}_{t-l'}\big] \\
&= \sum_{a'} \sum_{B'} |\sum_a \sum_B (P_t^\mu(B', a' \mid B, a) - \tilde{P}^\mu(B', a' \mid B, a)) \cdot \big(\mathbb{I}[(B_{t-1}, a_{t-1}) = (B, a)]r(B, a)\big)| \\
&\leq 2\bar{r}\|P_t^\mu - \tilde{P}^\mu\|_{TV} \\
&\leq \bar{r} \sum_{a'} \sum_{B'} |\mu(a \mid B) \sum_{y,s,s'} \big(\mathbb{I}[\mathcal{G}(B \parallel \{y,a\} = B']\Phi(y \mid s')\big) \cdot \big(\mathcal{P}(s' \mid s,a)(\pi(s \mid B) - \pi(s \mid H_t))\big)| \\
&= \bar{r} \sum_{B'} |\sum_{y,s,s'} \big(\mathbb{I}[\mathcal{G}(B \parallel \{y,a\} = B']\Phi(y \mid s')\big) \cdot \big(\mathcal{P}(s' \mid s,a)(\pi(s \mid B) - \pi(s \mid H_t))\big)| \\
&\leq 2\bar{r}(1-\rho)^l,
\end{aligned}
$$

where the last inequality holds using Lemma 3. Therefore, solving Eq. (23) recursively, we get

$$
\mathbb{E}\big[\mathbb{E}\big[\big(\Phi^T(\theta_{t-l'} - \hat{\theta})\big)^T \big(D_t \tilde{r} - \tilde{D}^\mu \tilde{r} \mid \mathcal{F}_{t-l'}\big)\big]\big] \leq \bar{r}R\Big(2(1-\rho)^l + 2\frac{(1-\rho)^l}{\rho'} + (1-\rho')^{l'}\Big) \, \forall t.
$$

Similarly, since $\|\Phi^T \theta_{t-l}\|_\infty \le R$, we can bound the sum of the last two terms in (22) as

$$\mathbb{E}\big[\mathbb{E}\big[(\Phi^T(\theta_{t-l'} - \hat\theta))^T(\gamma E_t \Phi^T \theta_{t-l'} - \gamma \tilde{D}^\mu \tilde{P}^\mu \Phi^T \theta_{t-l'}) \mid \mathcal{F}_{t-l'}\big]\big]$$

$$+ \mathbb{E}\big[\mathbb{E}\big[(\Phi^T(\theta_{t-l'} - \hat\theta))^T(\tilde{D}^\mu \Phi^T \theta_{t-l'} - D_t \Phi^T \theta_{t-l'}) \mid \mathcal{F}_{t-l'}\big]\big]$$

$$\le (1 + \gamma(1 - \rho'))R\Big((1 - \rho')^{l'} + 2\frac{(1 - \rho)^l}{\rho'}\Big).$$

Therefore, putting everything together, for a constant stepsize $\epsilon$, we have

$$\mathbb{E}[\mathcal{L}(\theta_{t+1})] \le (1 - 2\epsilon + 2\epsilon\gamma)\mathbb{E}[\mathcal{L}(\theta_t)] + \epsilon^2(\bar r + 2R)^2 + 2\epsilon R\bar r(1 - \rho)^l$$

$$+ 2\epsilon(R\bar r + R^2(1 + (1 - \rho')\gamma))\Big((1 - \rho')^{l'} + 2\frac{(1 - \rho)^l}{\rho'}\Big) + 2C_2(\bar r + 2R)l'\epsilon^2.$$

Using the above relation recursively, we have

$$\mathbb{E}\|\theta_{\tau+l'} - \hat\theta\|_2 \le (1 - 2\epsilon + 2\epsilon\gamma)^\tau\|\theta_{l'} - \hat\theta\|_2 + \Big[\frac{1 - (1 - 2\epsilon + 2\epsilon\gamma)^\tau}{2\epsilon(1 - \gamma)}\Big] \cdot \Big[\epsilon^2(\bar r + 2R)^2 + 4\epsilon R\bar r(1 - \rho)^l$$

$$+ 2C_2(\bar r + 2R)l'\epsilon^2 + 2\epsilon(R\bar r + R^2(1 + (1 - \rho')\gamma))\Big((1 - \rho')^{l'} + 2\frac{(1 - \rho)^l}{\rho'}\Big)\Big].$$

Therefore,

$$\mathbb{E}\|\bar{Q}^\mu_{\tau+l'} - \tilde{Q}^\mu\|_\infty \le \|\Phi\hat\theta - \tilde{Q}^\mu\|_\infty + (1 - 2\epsilon + 2\epsilon\gamma)^\tau\|\theta_{l'} - \hat\theta\|_2$$

$$+ \Big[\frac{1 - (1 - 2\epsilon + 2\epsilon\gamma)^\tau}{2\epsilon(1 - \gamma)}\Big] \cdot \Big[\epsilon^2(\bar r + 2R)^2 + 4C_2(\bar r + 2R)l'\epsilon^2 + 4\epsilon R\bar r(1 - \rho)^l$$

$$+ 2\epsilon(R\bar r + R^2(1 + (1 - \rho')\gamma))\Big((1 - \rho')^{l'} + 2\frac{(1 - \rho)^l}{\rho'}\Big)\Big].$$

Finally, by choosing $\epsilon = \frac{1}{\sqrt{\tau}}$ and $l' = \frac{\log \tau}{2\log(1-\rho')}$ for sufficiently large $\tau$ such that $2\epsilon(1 - \gamma) < 1$, we get

$$\mathbb{E}\|\bar{Q}^\mu_{\tau+l'} - \tilde{Q}^\mu\|_\infty \le \|\Phi\hat\theta - \tilde{Q}^\mu\|_\infty + (1 - \frac{2(1 - \gamma)}{\sqrt{\tau}})^\tau\|\theta_l - \hat\theta\|_2$$

$$+ \Big[\frac{1 - (1 - 2(1 - \gamma)/\sqrt{\tau})^\tau}{(1 - \gamma)}\Big] \cdot \Big[\frac{(\bar r + 2R)^2}{2\sqrt{\tau}} + \frac{C_2(\bar r + 2R)\log \tau}{\log(1 - \rho')\sqrt{\tau}} + 2R\bar r(1 - \rho)^l$$

$$+ \frac{2}{\rho'}(1 - \rho)^l(R\bar r + R^2(1 + (1 - \rho)\gamma)) + (1 - \rho')^{\frac{\log \tau}{2\log(1-\rho')}}(R\bar r + R^2(1 + (1 - \rho)\gamma))\Big]$$

$$:= \xi_{\text{TD-Error}}$$

### A.7  Proof of Theorem 3

Before we prove the regret bound for our policy iteration algorithm, we first mention an important result from the online learning literature, which will be used to derive the result.

Consider a game between two players consisting of $M$ rounds. At the beginning of each round $i$, the environment chooses a loss function $l_i : \mathcal{A} \to [0, 1]$, while the learner selects an action $a_i \in \mathcal{A}$. After both choices are made, the learner observes the loss $l_i(a_i)$, while the environment observes $a_i$. The learner's goal is to minimize its regret with respect to a fixed action $a^*$, which is defined as

$$\bar{\mathcal{R}}_M = \sum_{i=1}^{M} (l_i(a_i) - l_i(a^*)).$$

The following lemma provides a high probability bound on $\bar{\mathcal{R}}_M$.

**Lemma 6** *Cesa-Bianchi and Lugosi [2006] For the game mentioned above, assume that at round $i$ the learner chooses an action $a_i = a$ with probability $\mu_i(a) \propto \exp(-\eta \sum_{j=1}^{i-1} l_j(a))$, where $\eta = \sqrt{8 \log |\mathcal{A}|/M}$. Moreover, let $\delta \in (0, 1)$ and $a^* \in \mathcal{A}$ be an arbitrary fixed action. Then regardless of how the environment plays, with probability at least $1 - \delta$, we have*

$$\bar{\mathcal{R}}_M \le \sqrt{M \log |\mathcal{A}|/2} + \sqrt{M \log(1/\delta)/2}.$$

We will now use this result to prove the regret in Theorem 3. Using the definition of regret, we have

$$
\begin{aligned}
\mathcal{R}_T &= (\tau + l') \sum_{i=1}^{M} \mathbb{E}\Big[V^{\mu^*}(\boldsymbol{\pi}(H_0)) - \tilde{V}^{\mu_i}(\mathcal{G}(H_0))\Big] \\
&= (\tau + l') \sum_{i=1}^{M} \mathbb{E}\Big[V^{\mu^*}(\boldsymbol{\pi}(H_0)) - \tilde{V}(\mathcal{G}(H_0))\Big] + (\tau + l') \sum_{i=1}^{M} \mathbb{E}\Big[\tilde{V}(\mathcal{G}(H_0)) - \tilde{V}^{\mu_i}(\mathcal{G}(H_0))\Big] \\
&\leq M(\tau + l')\xi_{\text{POMDP}}^{\text{SMDP}} + (\tau + l') \sum_{i=1}^{M} \mathbb{E}\Big[\tilde{V}(\mathcal{G}(H_0)) - \tilde{V}^{\mu_i}(\mathcal{G}(H_0))\Big] \\
&\qquad,
\end{aligned}
\tag{24}
$$

where the inequality is obtained using Theorem 2. Therefore, using the Performance Difference Lemma Kakade and Langford [2002], for any $i = 1, \ldots, M$, we have

$$
\begin{aligned}
\mathbb{E}\big[\tilde{V}(\mathcal{G}(H_0)) - \tilde{V}^{\mu_i}(\mathcal{G}(H_0))\big] &= \mathbb{E}\Big[\mathbb{E}_{a\sim\tilde{\mu},a'\sim\mu_i}\big[\tilde{Q}^{\mu_i}(\mathcal{G}(H_0),a) - \tilde{Q}^{\mu_i}(\mathcal{G}(H_0),a')\big]\Big] \\
&= \mathbb{E}\Big[\mathbb{E}_{a\sim\tilde{\mu},a'\sim\mu_i}\big[\bar{Q}_{\tau+l'}^{\mu_i}(\mathcal{G}(H_0),a) - \bar{Q}_{\tau+l'}^{\mu_i}(\mathcal{G}(H_0),a')\big]\Big] \\
&\quad + \mathbb{E}\Big[\mathbb{E}_{a\sim\tilde{\mu},a'\sim\mu_i}\big[\tilde{Q}^{\mu_i}(\mathcal{G}(H_0),a) - \bar{Q}_{\tau+l'}^{\mu_i}(\mathcal{G}(H_0),a)\big]\Big] \\
&\quad + \mathbb{E}\Big[\mathbb{E}_{a\sim\tilde{\mu},a'\sim\mu_i}\big[\bar{Q}_{\tau+l'}^{\mu_i}(\mathcal{G}(H_0),a') - \tilde{Q}^{\mu_i}(\mathcal{G}(H_0),a')\big]\Big].
\end{aligned}
$$

Next, using Lemma 4, we know that $\mathbb{E}[\|\bar{Q}_{\tau+l'}^{\mu_i} - \tilde{Q}^{\mu_i}\|_\infty] \leq \xi_{\text{TD-error}} \; \forall i$. Therefore, we can write

$$
\begin{aligned}
&(\tau + l') \sum_{i=1}^{M} \mathbb{E}\Big[\tilde{V}(\mathcal{G}(H_0)) - \tilde{V}^{\mu_i}(\mathcal{G}(H_0))\Big] \\
&= (\tau + l') \sum_{i=1}^{M} \mathbb{E}\Big[\mathbb{E}_{a\sim\tilde{\mu},a'\sim\mu_i}\big[\bar{Q}_{\tau+l'}^{\mu_i}(\mathcal{G}(H_0),a) - \bar{Q}_{\tau+l'}^{\mu_i}(\mathcal{G}(H_0),a')\big]\Big] + 2(\tau+l')M\xi_{\text{TD-error}} \\
&= (\tau + l')\mathbb{E}\Big[\sum_{i=1}^{M}\Big(\langle \mu_i(\cdot \mid \mathcal{G}(H_0)), \bar{Q}_{\tau+l'}^{\mu_i}(\mathcal{G}(H_0),\cdot)\rangle - \langle \tilde{\mu}(\cdot \mid \mathcal{G}(H_0)), \bar{Q}_{\tau+l'}^{\mu_i}(\mathcal{G}(H_0),\cdot)\rangle\Big)\Big] \\
&\quad + 2M(\tau + l')\xi_{\text{TD-error}},
\end{aligned}
\tag{25}
$$

where the last equality follows from the linearity of expectation and expanding the inner expectation. Now, we can apply Lemma 6 to upper-bound (25). To this end, we can think of a game between the adversary and a player, which occurs over $M$ rounds. In each round $i$, the adversary chooses the loss function $l_i(\cdot) := \bar{Q}_{\tau+l'}^{\mu_i}(\mathcal{G}(H_0),\cdot)$, [13] and the player selects an action $a_i \in \mathcal{A}$ with probability $\mu_i(a_i|\mathcal{G}(H_0))$, which due to the structure of the policy updates in Algorithm 2 follows the same exponential update rule as in Lemma 6. Moreover, since any MDP (and in particular the Superstate MDP) admits a deterministic stationary policy Bertsekas and Tsitsiklis [1996], it follows that $\tilde{\mu}$ is a deterministic policy that always selects an optimal fixed action $a^*$, i.e.,

$$
\tilde{\mu}(a^*|\mathcal{G}(H_0)) = 1 \quad \text{and} \quad \tilde{\mu}(a|\mathcal{G}(H_0)) = 0 \quad \forall a \neq a^*.
$$

By applying Lemma 6, we conclude that the expectation of the inner product in (25) is upper-bounded by

$$
R\left(M\delta + \sqrt{\frac{M\log|\mathcal{A}|}{2}} + \sqrt{\frac{M\log(1/\delta)}{2}}\right).
$$

Substituting this bound into (25) and combining it with (24), we obtain

---

[13]The Q-function is upper-bounded by $R$ since $\|\theta\|_2 \leq R$ and $\|\phi(B,a)\| \leq 1$. This only scales the regret bound in Lemma 6 by a factor of $R$.

$$\mathcal{R}_T \leq (\tau + l')R\left(M\delta + \sqrt{\frac{M\log|\mathcal{A}|}{2}} + \sqrt{\frac{M\log(1/\delta)}{2}}\right) + M(\tau + l')\left(2\xi_{\text{TD-error}} + \xi_{\text{POMDP}}^{\text{SMDP}}\right)$$

$$= (\tau + l')R\left(\sqrt{M} + \sqrt{\frac{M\log|\mathcal{A}|}{2}} + \sqrt{\frac{M\log M}{4}}\right) + T\left(2\xi_{\text{TD-error}} + \xi_{\text{POMDP}}^{\text{SMDP}}\right),$$

where in the second step, we have chosen $\delta = 1/\sqrt{M}$ and used the fact that $M(\tau + l') = T$. Finally, by choosing $\tau + l' = \sqrt{T}$ and substituting the values for $\xi_{\text{TD-error}}$ and $\xi_{\text{POMDP}}^{\text{SMDP}}$, we obtain the desired result.

### A.8 Improving Bounds in Subramanian and Mahajan [2019] using Lemma 2

We state an important lemma from Subramanian and Mahajan [2019]. For additional details, the reader is referred to Subramanian and Mahajan [2019]. Before stating the lemma, we first present the definition of the approximate information state, as defined in Subramanian and Mahajan [2019]. Note that we have simplified the definition for the case of POMDPs.

**Definition 2** *Define $z \in \mathbb{Z}$ to be an approximate information state and $\sigma : \mathbb{H} \to \mathbb{Z}$ to be the approximate information state generator. Further, define $\hat{r} : \mathbb{Z} \times \mathcal{A} \to \mathbb{R}$ to be the reward approximation function. Additionally, let $(\epsilon, \delta)$ be fixed constants. Then $Z_t = \sigma(H_t)$ satisfies the following properties:*

- *For any history $H_t$ and action $a_t$, we have*

$$|\mathbb{E}[R_t \mid H_t = h_t, A_t = a_t] - \hat{r}(\sigma(H_t), a_t)| \leq \epsilon.$$

- *For any history $H_t$ and action $a_t$ and any Borel subset $B \in \mathbb{Z}$, define $\mu_t(B) = \hat{P}(Z_{t+1} \in B \mid H_t = h_t, A_t = a_t)$ and $v_t(B) = \mathbb{P}(B \mid \sigma(H_t), a_t)$. Then*

$$d_{TV}(\mu_t, v_t) \leq \delta.$$

Note that the grouping operator $\mathcal{G}$ in our work corresponds to the approximate information state generator $\sigma$ in their paper.

**Lemma 7** *Consider the approximate information state generator $\sigma$ to be a function which takes a history $H \in \mathbb{H}$ to an approximate information state $z \in \mathbb{Z}$, where for simplicity of our analysis, assume $\mathbb{Z}$ is a discrete set. Define a fixed point equation for the approximate information states as follows:*

$$\hat{V}(z, a) = \max_a \left[\hat{r}(z, a) + \gamma \sum_{\mathbb{Z}} \hat{V}(z')\hat{P}(z' \mid z, a)\right].$$

*Let $\hat{Q}^*$ denote the solution of the fixed point equation and $Q^*$ denote the optimal Q-function for the POMDP. Then, for all histories $H_t \in \mathbb{H}$, using results from Lemma 49 and Theorem 27 in Subramanian and Mahajan [2019], we have*

$$|\hat{V}^*(H_t) - V^*(\sigma(H_t))| \leq \frac{\epsilon}{1 - \gamma} + \frac{2\gamma\delta\bar{r}}{(1 - \gamma)^2}.$$

In the following, we show how to modify the proof in Subramanian and Mahajan [2019] to obtain a better bound using Lemma 2.

**Proof:**

$$|\hat{V}^*(H_t) - V^*(\sigma(H_t))|$$

$$= |\max_a \left[\mathbb{E}[R_t + \gamma V^*(H_{t+1}) \mid H_t = h_t, A_t = a_t]\right] - \max_a \left[\hat{r}(\sigma(h_t), a_t) + \gamma \sum_{z'} \hat{V}(z')\hat{P}(z' \mid z, a)\right]|$$

$$\leq |\mathbb{E}[R_t \mid H_t = h_t, A_t = a'] - \hat{r}(\sigma(h_t), a')| + \gamma|\mathbb{E}[V^*(H_{t+1}) \mid H_t = h_t, A_t = a'] - \sum_{z'} \hat{V}(z')\hat{P}(z' \mid z, a')|$$

$$\leq \epsilon + \gamma| \sum_y V^*(h_t \parallel \{y, a'\})\mathbb{P}(y \mid H_t = h_t, A_t = a') - \sum_{z'} \hat{V}(z')\hat{P}(z' \mid \sigma(h_t), a')|$$

$$= \epsilon + \gamma| \sum_{z'} \Big( \sum_{y:\sigma(h_t \parallel \{y,a'\})=z'} V^*(h_t \parallel \{y, a'\}) \frac{\mathbb{P}(y \mid H_t = h_t, A_t = a')}{\sum_{y:\sigma(h_t \parallel \{y,a'\})=z'} \mathbb{P}(y \mid H_t = h_t, A_t = a')} \Big) \hat{P}(z' \mid H_t, a')$$

$$- \sum_{z'} \hat{V}(z')\hat{P}(z' \mid \sigma(h_t),' a)| \tag{26}$$

Suppose, for all $H_t$, $|\hat{V}^*(H_t) - V^*(\sigma(H_t))| \leq \omega$. This implies that

$$\Big| \sum_{y:\sigma(h_t \parallel \{y,a'\})=z'} V^*(h_t \parallel \{y, a'\}) \frac{\mathbb{P}(y \mid H_t = h_t, A_t = a')}{\sum_{y:\sigma(h_t \parallel \{y,a'\})=z'} \mathbb{P}(y \mid H_t = h_t, A_t = a')} - \sum_{y:\sigma(h_t \parallel \{y,a'\})=z'} \tilde{V}(z') \Big| \leq \omega \tag{27}$$

At this point, we can use Lemma 2. Therefore, we obtain

$$\omega \leq \epsilon + \gamma \frac{\delta\bar{r}}{(1-\gamma)} + \gamma\omega\Big(1 - \frac{\delta}{2}\Big)$$

$$\leq \frac{\epsilon}{1 - \gamma + \delta/4} + \frac{\gamma\delta\bar{r}}{(1-\gamma)(1-\gamma+\delta/2)}. \tag{28}$$

### A.9 Comparison of our bounds in Theorem 2 with Cayci et al. [2024]

We can do a direct comparison with Theorem 4.4 of Cayci et al. [2024] and show how we obtained our improved bound. Specifically, our Theorem 3 aggregates the bound from our Theorem 2, the TD-learning error from Lemma 4, and the final error due to policy optimization. Each component can be compared as follows:

1. Error due to the mismatch between optimal value functions of the POMDP and SuperState MDP — corresponds to Theorem 2 and is comparable to $\epsilon_{inf}$ in Cayci et al. [2024].

2. Standard TD-learning error — the terms in Lemma 4 without $(1-\rho)^l$, comparable to the first term of $\epsilon_{critic}$ in Cayci et al. [2024].

3. Additional TD-learning error from sampling mismatch, i.e., the terms in Lemma 4 with $(1-\rho)^l$ (comparable to $\epsilon_{pa}$ of Cayci et al. [2024].

4. Function approximation error — similar to $l_{CFA}$ in Cayci et al. [2024].

5. Policy optimization error - comparable to $\epsilon_{actor}$ in Cayci et al. [2024].

We improve upon the bounds in (1) and (3) which leads to the difference mentioned at the beginning of this response. We believe our analysis offers a detailed decomposition and sharp insight into the sources of error.

## B   Analysis of the Practical Validity of Assumption 1

### B.1   Analyzing Assumption 1 for a Practical Example

We consider a practical example of modelling **Customer Behavior Modeling in Retail**.

Here:

States represent engagement levels such as {Uninterested, Browsing, Considering, Purchasing}.

Observations are features like the number and type of clicks (e.g., "Viewed Product", "Added to Cart").

In such systems, observations from adjacent engagement states tend to overlap significantly: for instance, both "Browsing" and "Considering" may involve product views and occasional cart additions. Furthermore, customer behavior is highly dynamic—users frequently move between these states in short time spans (e.g., from "Considering" back to "Browsing" or forward to "Purchasing"). This overlap in observation distributions and frequent transitions among states leads to high mixing, thereby increasing the Dobrushin coefficient and ensuring filter stability.

Next we present a simplified example to model customer behaviour:

**States:** $s_0$ (Uninterested), $s_1$ (Browsing), $s_2$ (Considering), $s_3$ (Purchasing).

**Actions:** $a_0$: Show generic homepage, $a_1$: Recommend trending products

**Observations:** $y_0$: No clicks, $y_1$: Viewed product, $y_2$: Added to cart, $y_3$: Purchased

**Transition Probabilities for $a_0$:**

| $s_t \rightarrow s_{t+1}$ | $s_0$ | $s_1$ | $s_2$ | $s_3$ |
|---|---|---|---|---|
| $s_0$ | 0.4 | 0.4 | 0.1 | 0.1 |
| $s_1$ | 0.3 | 0.3 | 0.2 | 0.2 |
| $s_2$ | 0.2 | 0.3 | 0.3 | 0.2 |
| $s_3$ | 0.1 | 0.2 | 0.4 | 0.3 |

**Transition Probabilities for $a_1$:**

| $s_t \rightarrow s_{t+1}$ | $s_0$ | $s_1$ | $s_2$ | $s_3$ |
|---|---|---|---|---|
| $s_0$ | 0.4 | 0.3 | 0.2 | 0.1 |
| $s_1$ | 0.2 | 0.4 | 0.2 | 0.2 |
| $s_2$ | 0.1 | 0.3 | 0.4 | 0.2 |
| $s_3$ | 0.1 | 0.2 | 0.3 | 0.4 |

**Observation Kernel:**

| $s_t \rightarrow y_t$ | $y_0$ | $y_1$ | $y_2$ | $y_3$ |
|---|---|---|---|---|
| $s_0$ | 0.8 | 0.2 | 0.0 | 0.0 |
| $s_1$ | 0.3 | 0.5 | 0.2 | 0.0 |
| $s_2$ | 0.1 | 0.3 | 0.4 | 0.2 |
| $s_3$ | 0.0 | 0.1 | 0.3 | 0.6 |

Therefore, referring to Theorem 5 of Kara and Yuksel [2020], we calculate the Dobrushian coeffients to be $\delta(\mathcal{P}) = 0.5$ and $\delta(\Phi) = 0.1$, which satisfies the sufficient condition:

$$(1 - \delta(\mathcal{P}))(1 - \delta(\Phi)) < 1,$$

implying that the Filter Stability condition is satisfied for this case.

We believe that Assumption 1 covers many practical examples and future work can consider a multi-step variant where the system exhibits contraction after every $k$ steps. We believe that our results can be extended to such a setting.

## B.2 A Counter Example

To motivate future work, we also provide a counter example (suggested by one of the reviewers) where Assumption 1 does not hold.

Consider a T-Maze as described in Bakker [2001]. Here we present an infinite-horizon variant. The domain consists of a T-shaped maze, as shown in Fig 1. The agent is initially located at the beginning of a corridor (position S), which is followed by a junction (position X) where the agent can step into two different directions, and can keep going into that direction. The goal of the agent is to step into

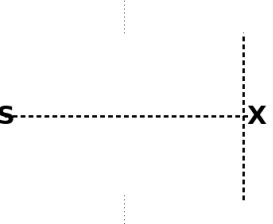

Figure 1: T-Maze

the correct direction, where the correct direction is determined by a piece of information the agent is given at the very beginning, when he is located at the beginning of the corridor (position S). The agent is given a non-zero reward $R > 0$ in every cell after taking the correct direction.

The T-maze is naturally modelled as a POMDP. The hidden states consist of

- An initial state $s_0$

- States $s_i^d$ for every position in the corridor, including the junction, standing for the fact that the agent is in position $i$ and the initial observation specified that the correct direction is $d \in \{1, 2\}$ (consider that $d = 1$ stands for "left" and $d = 2$ stands for right).

- States $r_i^d$ for $i \in \mathbb{N}$ and $d \in \{1, 2\}$, standing for the fact that the agent is in the $i^{th}$ cell after stepping into the first of the two possible directions, and given that the initial observation said that the correct direction to take is direction $d$

- States $q_i^d$ for $i \in \mathbb{N}$ and $d \in \{1, 2\}$, standing for the fact that the agent is in the $i^{th}$ cell after stepping into the second of the two possible directions, and given that the initial observation said that the correct direction to take is direction $d$.

  Note that transitions in the hidden state space are deterministic. Given the entire history of observations and actions, the agent can determine the current hidden state, hence belief states are sharp, always assigning probability 1 to a hidden state and 0 to the others. If only a suffix of the history is available, the agent can only determine its current location, and it cannot distinguish between states $s_i^1$ and $s_i^2$, between states $r_i^1$ and $r_i^2$, and between states $q_i^1$ and $q_i^2$. In other words, belief states will assign probability 1/2 to each of the two possible states.

  The optimal value function, given an entire history $H$ is roughly $V^* = R(1/(1 - \gamma))$. However, the value of any superstate $\mathcal{G}(H)$ is roughly $V^*/2$, since the value is determined by the belief, which is uniform over non-distinguishable stats once we remove the first observation. The difference between the two values is $V^*/2$, which is constant wrt to $l$ as opposed to the bound which we obtain in Theorem 2 which goes to 0 for $l \to \infty$. The reason why Theorem 2 does not hold in this case is due to the contraction property required for Assumption 1 being non-applicable due to the unique structure of the POMDP.

## C  Experimental Results

### C.1  Effect of History Length and Observation Noise

Previous works in the literature have focused heavily on solving POMDPs by considering past $k$ observations to design optimal policy. However, for the sake of completeness, we evaluate the performance of our algorithm on a partially observable variant of the FrozenLake-v1 environment. The environment is modified to introduce observation noise, where with probability $p$, the agent receives a random observation different from the true state. All the implementation code is available at this code repository: https://github.com/ameyanjarlekar/Policy-Based-RL-For-POMDPs.

The agent uses a history-based state representation by maintaining the last $k$ observation-action pairs. We study the impact of both the observation noise probability $p$ and the history length $k$ on the agent's performance.

**Setup.**

- Environment: FrozenLake-v1 with `is_slippery=False`.
- Observation Noise: With probability $p$, the observed state is replaced with a random incorrect state.
- History: The agent encodes the last $k$ (observation, action) pairs as the state.
- Algorithm: POLITEX with TD(0)-based Q-value estimation and exponentiated gradient policy updates.
- Compute: All the experiments were performed on the Google Colab CPU.

**Results.** The following plot captures the average reward per episode for varying history lengths $k$ and observation noise levels $p$. The moving average was computed over a sliding window of episodes.

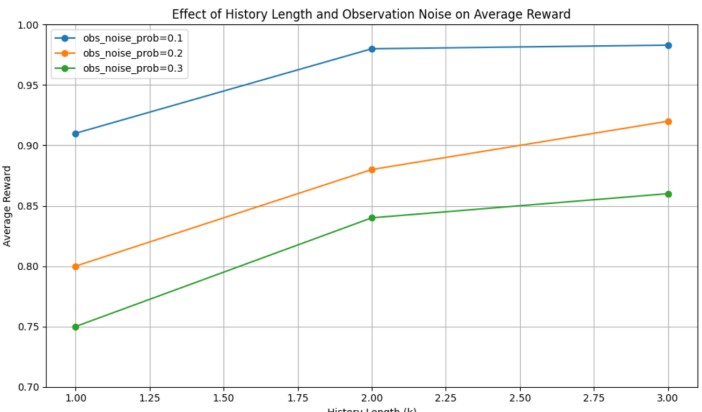

Figure 2: Average reward per episode for varying history lengths $k$ and observation noise levels $p$.

**Observations.**

- Increasing the history length $k$ consistently improves performance under partial observability.
- Higher observation noise degrades performance, but the use of longer histories mitigates the effect.
- Even with moderate to high noise levels ($p = 0.3$), using $k = 3$ allows the agent to recover much of the performance.

### C.2 Comparison with Cayci et al. [2024]

In our paper, we show that we improve upon the bound in Cayci et al. [2024]. using a computationally lighter algorithm. To experimentally illustrate this result, we consider an example for a simple partially observable Markov decision process (POMDP) with two states, two actions, and two observations. The environment is stochastic, with state transitions and observations defined by fixed probabilities, and rewards designed to encourage taking the correct action in the hidden state.

To handle partial observability, we represent the agent's state by a finite history of recent action-observation pairs, with history lengths of 1 and 2 tested.

For each setting, the agent trains over 200 episodes, each of fixed length 20 steps. The learning rate $\alpha$ is set to 0.1 and the discount factor $\gamma$ to 0.9. Policies are represented tabularly and updated greedily with respect to Q-values after each episode.

We measure the agent's performance by total reward accumulated per episode and analyze learning curves as well as final average rewards (averaged over the last 20 episodes) to assess convergence and policy quality.

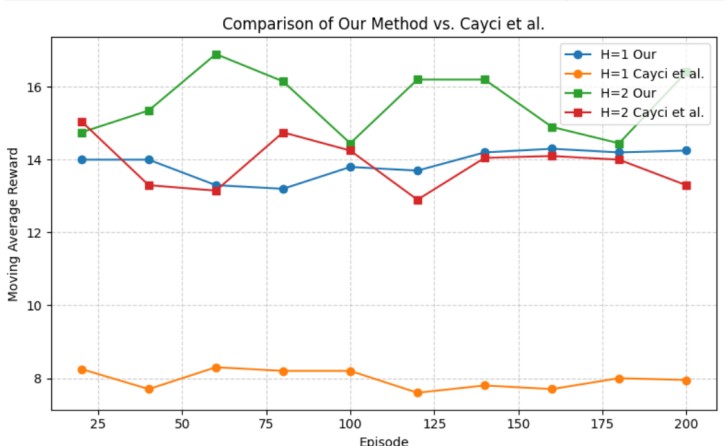

Figure 3: Comparing Moving Average Reward

Additionally, several prior works have empirically demonstrated the effectiveness of using finite histories of observations as surrogates for hidden states in partially observable reinforcement learning problems. For example, Paischer et al. [2022] leverages pretrained language transformers to compress observation histories into compact representations, showing improved sample efficiency on POMDP benchmarks. This is similar to our work if we consider the feature vectors generated by the language transformers as the feature vectors. Similarly, Li et al. [2024] presents a framework that adaptively uses privileged state information during training while deploying policies that rely on observation histories. Other related studies, such as Ni et al. [2022] and Aberdeen et al. [2007], support the use of history-based or memory-augmented policies with policy gradient methods.

