# OpenReview forum: "Scalable Policy-Based RL Algorithms for POMDPs"
_NeurIPS.cc/2025/Conference — NeurIPS 2025 poster_

### Official Review · Reviewer_2QJ5 · 2025-06-21

**Clarity:** 3
**Significance:** 3
**Originality:** 3
**Rating:** 4
**Confidence:** 4

**Summary:**

This paper addresses the challenges of solving partially observable reinforcement learning (PORL) problems, where the continuous belief states in POMDPs complicate optimal policy learning. The authors propose approximating the POMDP with a finite-state Markov Decision Process (MDP), referred to as a Superstate MDP, constructed from finite historical states. They establish theoretical guarantees that explicitly link the optimal value function of this simplified Superstate MDP to that of the original POMDP. Leveraging this framework, the authors introduce a policy-based learning approach utilizing linear function approximation, combining TD-learning and policy optimization to solve the approximated problem. They show that the approximation error decreases exponentially with the history length. Notably, this work provides the first finite-time error bounds quantifying the inaccuracies that arise when applying standard TD-learning methods to problems with non-Markovian dynamics.

**Questions:**

(Q1) Can you provide a comparison table of the current result with those from preexisting work, e.g. Subramanian and Mahajan 2019, Cayci et al. (2024), etc.

(Q2) Can you provide empirical study result comparing your proposed algorithm with those preexisting ones referred, e.g. Cayci (2024)?

(Q3) The paper builds upon the superstate MDP construction introduced by Kara and Yuksel (2023). How is the theoretical analysis presented here differs from Kara and Yuksel (2023)?

(Q4) A related but distinct paper [1] explores convergence properties of TD learning in Markov Reward Processes incorporating hidden state information. Could the authors clarify the connections to and differences from their current framework and theoretical results?

[1] Amiri, M., & Magnússon, S. (2024, June). On the Convergence of TD-Learning on Markov Reward Processes with Hidden States. In 2024 European Control Conference (ECC) (pp. 2097-2104). IEEE.

**Ethical Concerns:**

["NO or VERY MINOR ethics concerns only"]

**Final Justification:**

I appreciate the author's detailed response and I'll maintain my score.

**Limitations:**

Yes

**Quality:**

3

**Strengths And Weaknesses:**

Strength:
The paper is clearly written and effectively identifies the research gap along with relevant literature. It is technically rigorous, and its the first to provide convergence guarantees for TD-learning within the POMDP framework.

Weakness:
please refer to questions in the below.

---

> ### Author Rebuttal · Authors · 2025-07-30
>
> We sincerely thank the reviewer for their thoughtful comments and valuable feedback, which have helped us improve the quality and clarity of the paper. We hope that the following clarifications help address the reviewer's concerns.
>
> ## Comparison Table with Recent Work
>
> We present below a comparison table of our results with preexisting works, namely, Kara et al., Mahajan et al and Cayci et al.
>
> | Aspect                           | Our Work                                                              | Mahajan et al.                                   | Cayci et al.                             | Kara et al.                                                               |
> | :------------------------------- | :-------------------------------------------------------------------- | :----------------------------------------------- | :---------------------------------------------- | :------------------------------------------------------------------------ |
> | **Value function gap bound** | Tighter bound; improves prior results with worst-case guarantee       | $ O\left( \frac{1}{(1-\gamma)^2} \right) $      | Polynomial in $ \frac{1}{1-\gamma} $            | $ O\left( \frac{1}{(1-\gamma)^2} \right) $ expected-case only             |
> | **Policy Optimization (PO) analysis** | First to show convergence for standard PO (TD-learning + policy update), no added complexity | Empirical deep RL; misaligned objective; no guarantees | Uses $ m $-step TD; high computational complexity | No PO analysis (Analyses Q-learning)                                      |
> | **Function approximation** | Finite-time bounds for PO with linear approximation                   | Not analyzed                                     | Analyzed but worse bounds                       | Do not analyze function approximation, also difficult to prove function approximation for Q-learning |                                                                  |
> | **Convergence guarantees** | Finite-time, non-asymptotic, under standard assumptions               | No theoretical convergence guarantees            | Requires $ m $-step TD; more computationally heavy | Asymptotic only                                                           |
>
>
> ## Difference between the theoretical analysis presented in our paper and Kara and Yuksel (2023)
>
> The theoretical analysis in this paper builds upon the superstate MDP construction introduced by Kara and Yuksel (2023), beyond that we study completely different algorithms and therefore the analysis is also different. Specifically, they study Q-learning without function approximation whereas we study policy based methods with function approximation. Some additional details on the differences between the two papers are as follows:
>
> **Improved Approximation Guarantees:** To obtain the result in Theorem 2, Kara and Yuksel (2023) use standard decomposition and triangle inequality techniques to bound errors, which can be loose and conservative. However, we introduce Lemma 2, which helps us to get a tighter bound.
>
> **Policy Optimization Algorithm:**
> Kara and Yuksel (2023) focus on the Q-learning algorithm whereas our paper focuses on the TD-learning followed by Policy Optimization framework.
> Their analysis benefits from a stationary exploratory sampling policy, which simplifies theory. Our work analyzes a TD-learning based policy optimization approach where the sampling policy changes at every update, making the sampling non-stationary relative to the underlying POMDP states. Additionally, we derive finite-time bounds which is not the case in Kara and Yuksel (2023).
>
> **Obtaining Finite Time Performance Bounds** The approach in Kara and Yuksel (2023) is to express the algorithm updates in terms of iterates which allows one to use Robbins-Monro type algorithms, to prove asymptotic convergence. Since our focus is on getting a Finite Time Performance Bound, we use a completely different analysis where we use the contraction property of the Bellman operator along with other algebraic techniques to relate the error improvement between successive steps of the TD-learning updates. Additionally, we need to handle an additional sampling mismatch error due to the non-stationary nature of the sampling policy while deriving bounds for TD-learning.
>
> **Extending to Function Approximation Setting:** We extend the theoretical guarantees to linear function approximation settings in policy optimization, allowing scalability to larger or continuous state spaces. Kara and Yuksel (2023) do not address this.
>
> **Providing Regret Bounds:** Since the algorithm updates the policy at every step, a more accurate way of capturing the performance of the algorithm is through regret bounds, which is not explored by Kara and Yuksel (2023).
>
> ## Comparison to the work by Amiri and Magnusson
>
> Thank you for letting us know about this reference. However, based on our reading of the paper, their algorithm does not seem to apply to POMDPs. Specifically, according to Assumption 2 in their paper, the hidden states and the observations evolve independently of each other. On the other hand, a key feature of POMDPs is that observations provide information about the hidden states. The only link in their paper between the hidden states and observations is through the reward function.
>
> ## Empirical Validation
> Reviewer TqLW has also made a similar comment. Due to space limitations, we have provided our response in the **Additional Experiments** section of the Comments to Reviewer TqLW, which provides an additional experiment comparing our work with Cayci et al.

---

> > ### Comment · Reviewer_2QJ5 · 2025-08-08
> >
> > I truly appreciate the author's thorough response, and I will keep my score at 4.

---

### Official Review · Reviewer_jAwy · 2025-06-22

**Clarity:** 2
**Significance:** 2
**Originality:** 3
**Rating:** 4
**Confidence:** 3

**Summary:**

In this paper, the authors address the problem of solving finite state and finite actions partially observable reinforcement learning problems. Specifically, they derive theoretical bounds on the approximation capability of superstate MDPs. Superstate MDPs have been introduced in previous work and are MDPs over histories of observation and actions of a fixed length.

The authors’ results show tighter bounds than previous work in the approximation capabilities of superstate MDPs, with an exponentially decaying error with the length of the history considered. Their proof rely on a bound on a specific algebraic identity with probability mass functions.

They use the proven bounds to show convergence of a TD-learning value estimation algorithm with linear approximation. They then extend the result to a policy optimization method using such TD-learning method in combination with a policy update method. The policy uses a softmax over the estimated Q-values. They show that the regret of the resulting algorithm is bounded by $T + O(T^{3/4} \log T).$

**Questions:**

- How constraining is really the uniform filter stability condition in practice? Would standard POMDP benchmark problems satisfy it? Can you also provide some intuitions of its meaning?
- In the proof for theorem 2, l492, you dropped the infinity norm why? You are also assuming that it is the same $\hat{a}$ maximizing the r.h.s of the Bellman operator for $V^*$ and $\tile{V}$, it does not seem correct, can you explain?
- I did not follow the end of the proof of lemma 3, you are constructing the $coefficient$ and most of them are 0s, but what shows that some of them are not, and that those specific coefficient correspond to states that only differ in the first two elements?

**Ethical Concerns:**

["NO or VERY MINOR ethics concerns only"]

**Final Justification:**

The authors clarified most of the concerns I had on the proofs of the theorems in the rebuttal. Thus I increase my score to borderline accept. It is still borderline because assumption 1 is quite limiting and the empirical evaluation is also limited (even if improved during the rebuttal).

**Limitations:**

The analysis is limited to discrete state and actions and linear approximation. This is clearly stated by the authors.

**Paper Formatting Concerns:**

no concerns.

**Quality:**

2

**Strengths And Weaknesses:**

**Strengths**

- The problem of RL in partially observable environment is important and has many applications. Previous works have shown the strength of methods relying on finite histories empirically but, as acknowledged by the authors, there is no strong theoretical results to support them.

- The result from the author seem to improve on previous work by providing tighter bounds for the approximation between the finite-history MDP representing the underlying POMDP.

- The theoretical result of the regret analysis of the proposed method might be novel. The derived TD algorithm is not particularly novel, and consists of policy optimization with an augmented state that is the last l state+action pairs, actually common practice in deep RL but being formalized in this paper.

**Weaknesses**

1. Lack of discussion of the main assumption

Assumption 1 is the main cornerstone of the results but no insights are provided. I think the authors should provide some intuitions about whether or not this assumption is strict and more details on when it holds or not. I understand that the total variation between reachable next belief state is smaller than the total variation between two belief state?
The result on the approximation quality of the MDP is mostly valid because of assumption 1, but little is said about the validity of this assumption.

2. Weak empirical validation

One plot is provided in appendix for Frozen lake but no discussion of the result is provided. It is said that the resulting algorithm is more efficient than the previous work, but no direct comparison is provided. No comparison is provided against naïve baseline like standard Q-learning on an augmented state space stacking the history (no belief).
With POMDPs, the question of scalability of algorithms with the horizon is very important. I am surprised that it is not discussed much in the paper.

3. presentation

The proofs were difficult to follow, and I have doubts on some steps (see questions) which makes it hard to evaluate correctness.

-	For the proof of lemma 2, number the lines, and maybe add a few words explaining the step to make the review easier. The third step when you introduce the max was not obvious to me.
-	In the proof of lemma 3, there is a typo l517, and l530.  The proof is really hard to follow and I was not really able to understand if it was correct. There is a switch from general distributions d1 and d2, to distribution over beliefs halfway.

**Writing feedback**
-	The related work on POMDP and internal state representation is a redundant with the intro
-	Using $\pi$ for the belief state is a bit confusing since it is traditionally used for policies. I suggest using $b$ as many POMDP papers do. Then you can use $\pi$ instead of $\mu$ for the policy.

---

> ### Author Rebuttal · Authors · 2025-07-30
>
> We sincerely thank the reviewer for their thoughtful comments and valuable feedback, which have helped us improve the quality and clarity of the paper. We hope that the following clarifications help address the reviewer's concerns.
>
> ## Clarification Regarding Proof of Theorem 2
>
> There was a typo in the original proof for Theorem 2, where the infinity norm was not present in some steps, which might have led to your confusion. Due to the character limit, we are unable to provide the entire detailed proof. Therefore, we provide more detailed steps for the parts of the proof that were confusing to you. We will add more detailed steps to our proofs in the revised draft of the paper.
>
> Let $\delta$ be defined as follows,
>
> \begin{equation}
>      \delta = ||  V^*\big(\boldsymbol{\pi}(H) \big) - \tilde{V}\big(\mathcal{G}(H) \big) ||  _{\infty}
> \end{equation}
>
> Next, we upper bound $|   V^*\big(\boldsymbol{\pi}(H) \big) - \tilde{V}\big(\mathcal{G}(H) \big)|$ for all $H$. First we can write it as
>
> \begin{align}
> \Big|\max_a \Big[ \sum_s \pi(s \mid  H) r(s,a) + \gamma \sum_y V^*\big(\boldsymbol{\pi}(H \cup \{y,a\})\big) \sigma(\boldsymbol{\pi}(H),y,a)\Big]
>     - \max_a \Big[ \tilde{r}(B,a) + \gamma \sum_y \tilde{V}\big(\mathcal{G}\big(B \cup \{y,a\})\big)\sigma(\boldsymbol{\pi}(B),y,a)\Big] \Big | .
> \end{align}
>
> Assume WLOG that the first term is greater than the second term. Next, suppose $\hat{a}$ maximizes $\sum_s \pi(s \mid  H) r(s,a) + \gamma \sum_y V^*\big(\boldsymbol{\pi}(H \cup \{y,a\})\big) \sigma(\boldsymbol{\pi}(H),y,a)$.
>
> Then the above expression equals
>
> \begin{align}
>     \Big[ \sum_s \pi(s \mid  H) r(s,\hat{a}) + \gamma \sum_y V^*\big(\boldsymbol{\pi}(H \cup \{y,\hat{a}\})\big) \sigma(\boldsymbol{\pi}(H),y,\hat{a})\Big] - \max_a \Big[\tilde{r}(B,a) + \gamma \sum_y \tilde{V}\big(\mathcal{G}\big(B \cup \{y,a\})\big)\sigma(\boldsymbol{\pi}(B),y,a)\Big]
> \end{align}
>
> Finally, since the maximum value of the second term will be greater than evaluating the second term for $a=\hat{a}$, it is upper bounded as follows
>
> \begin{align}
> \leq \Big[ \sum_s \pi(s \mid  H) r(s,\hat{a}) + \gamma \sum_y V^*\big(\boldsymbol{\pi}(H \cup \{y,\hat{a}\})\big) \sigma(\boldsymbol{\pi}(H),y,\hat{a})\Big] - \Big[\tilde{r}(B,\hat{a}) + \gamma \sum_y \tilde{V}\big(\mathcal{G}\big(B \cup \{y,\hat{a}\})\big)\sigma(\boldsymbol{\pi}(B),y,\hat{a})\Big]
> \end{align}
>
> Then following same steps in the proof mentioned in the paper, we get
> \begin{align}
>     |   V^*\big(\boldsymbol{\pi}(H) \big) - \tilde{V}\big(\mathcal{G}(H) \big)| \leq 2(1-\rho)^l\bar{r} + \gamma \bigg[ \big\|   \sigma(\boldsymbol{\pi}(H),\cdot,\hat{a})- \sigma(\boldsymbol{\pi}(B),\cdot,\hat{a}) \big\|  _{TV}\bigg(\frac{\bar{r}}{1-\gamma} - \frac{\delta}{2}\bigg) + \delta\bigg].
> \end{align}
> Therefore,
> Case 1: If $\big(\frac{\bar{r}}{1-\gamma} - \frac{\delta}{2}\big) > 0$. Then, we have
> \begin{align}
>     \delta \leq  2(1-\rho)^l\bar{r} + \gamma \bigg[ \sup_H ||   \sigma(\boldsymbol{\pi}(H),\cdot,\hat{a})- \sigma(\boldsymbol{\pi}(B),\cdot,\hat{a}) ||  _{TV}\bigg(\frac{\bar{r}}{1-\gamma} - \frac{\delta}{2}\bigg) + \delta\bigg]
> \end{align}
>
> Or in the other case we have
>
> \begin{align}
>     \delta \leq  2(1-\rho)^l\bar{r} + \gamma \bigg[ \inf_H ||   \sigma(\boldsymbol{\pi}(H),\cdot,\hat{a})- \sigma(\boldsymbol{\pi}(B),\cdot,\hat{a}) ||_{TV} \bigg(\frac{\bar{r}}{1-\gamma} - \frac{\delta}{2}\bigg) + \delta\bigg]
> \end{align}
>
> Now, since, both $\inf_H ||   \sigma(\boldsymbol{\pi}(H),\cdot,\hat{a})- \sigma(\boldsymbol{\pi}(B),\cdot,\hat{a})  ||_{TV}$ and
>
> $\sup_H ||   \sigma(\boldsymbol{\pi}(H),\cdot,\hat{a})- \sigma(\boldsymbol{\pi}(B),\cdot,\hat{a}) ||_{TV}$ are upper bounded by $2(1-\rho)^l$, we get the mentioned result.
>
>
> ## Clarification Regarding Proof of Lemma 3
>
> Next, to clarify the question regarding Lemma 3, we first present a short proof sketch detailing steps involved in the proof.
>
> We first assume (we will construct such $\alpha_{i,j}$ in algorithm 3) that there exists $\alpha_{i,j} \geq 0$ and $\sum_{i,j} \alpha_{i,j} = || d_1 - d_2 ||_{TV}$ such that
>
> $ || \tilde{\mathcal{P}}^{\mu}d_1- \tilde{\mathcal{P}}^{\mu}d_2 ||_{TV} $
>
>  $ \leq \sum_{i,j} \alpha_{i,j} || \tilde{\mathcal{P}}^{\mu}(e_i-e_j)||_{TV}$
>
> Now, since $ || \tilde{\mathcal{P}}^{\mu}(e_i-e_j)||_{TV}$ is less than or equal to 1,
>
>  if we can prove that $|| \tilde{\mathcal{P}}^{\mu}(e_i-e_j)||_{TV} < 1$ for some pair of $i,j$ such that $\alpha_{i,j} > 0$, we are guaranteed a contraction
>
> Next, we show that if $e_i$ and $e_j$ are superstates which only differ in the first two elements, then for such $e_i, e_j$, $ || \tilde{\mathcal{P}}^{\mu}(e_i-e_j)||_{TV} < 1$
>
> Finally, in the constructed algorithm we show that $\alpha_{i,j} > 0$ for such pair of $e_i,e_j$. Now since $\Delta_i, \Delta_j \neq 0$ we need to prove that $\Delta_i, \Delta_j \neq 0 \implies \alpha_{i,j} > 0$. See that $\alpha_{i,j} = \min(\Delta_i, - \Delta_j)$.
>
> Therefore, since $\Delta_i, \Delta_j \neq 0$, $\alpha_{i,j} > 0$. Additionally, the steps $\Delta_i \leftarrow \Delta_i - \alpha_{i,j}$ and $\Delta_j \leftarrow \Delta_j + \alpha_{i,j}$ ensures either one of them goes to 0 and thus is removed from the surplus/deficit set ensuring that $\alpha_{i,j}$ is not modified further.
>
> ## Regarding Uniform Filter Stability Condition (Assumption 1)
> Assumption 1, i.e., the filter stability means that every new observation sufficiently informs the agent to reduce differences between any two prior beliefs, ensuring the belief state “forgets” initial uncertainty and the filtering process is well-behaved. Therefore, one intuitive condition when Assumption 1 holds is when **the belief update is less sensitive to the prior and more influenced by new observations and actions**. This means that the next action and observation dictate what the next belief will be more than the current belief of the system.
>
> Therefore, to ensure that the filter stability condition holds, we need
>
> 1) **Sufficiently Mixing State Transitions**: The transition kernel $\mathcal{P}(s'\mid s,a)$ should be mixing, meaning from any state s, there's a positive probability to reach many other states s' over time.
>
> 2) **Non-deterministic and Informative Observations**: The observation kernel $\phi(y\mid s)$ must be non-deterministic but sufficiently informative.
>
> To check whether the filter stability condition holds, one sufficient condition uses the Dobrushin Coefficients of the Transition kernel and the Observation Kernel.
>
> From Theorem 5 of [1], if $(1-\delta(P))(2-\delta(\phi)) < 1$, then the filter stability condition holds. Here $\delta(\mathcal{P})$ is the minimum Dobrushin coefficient of the Transition kernel across all possible actions and $\delta(\phi)$ is the Dobrushin coefficient of the Observation kernel.
>
> Let P be a row-stochastic matrix over a finite state space $\mathbb{S}$. The Dobrushin coefficient $\delta(P)$ is defined as: $\delta(P(.\mid.,a)) = \inf_{x, y \in \mathbb{S}} \sum_{z \in \mathbb{S}} \min( P(z\mid x,a), P(z \mid y,a)) $
>
> and
> $\delta(P) = \min_a \delta(P(.\mid.,a))$
>
> This quantifies the minimum overlap between any two rows of the matrix P.
>
> This quantity measures how similar the rows of the matrix are; larger values indicate more mixing and hence, stronger contraction properties. As a result, when there is sufficient mixing—i.e., when every state has a non-trivial probability of transitioning to several other states or when every observation can be generated from multiple underlying states—the system exhibits filter stability. This scenario is common in dynamic environments or environments that have noisy observations, where the belief update is less sensitive to the prior and more influenced by new observations.
>
> A practical example is **Customer Behavior Modeling in Retail**. Here:
>
> States represent engagement levels such as {Uninterested, Browsing, Considering, Purchasing}.
>
> Observations are features like the number and type of clicks (e.g., “Viewed Product”, “Added to Cart”).
>
> In such systems, observations from adjacent engagement states tend to overlap significantly: for instance, both “Browsing” and “Considering” may involve product views and occasional cart additions. Furthermore, customer behavior is highly dynamic—users frequently move between these states in short time spans (e.g., from "Considering" back to "Browsing" or forward to "Purchasing"). This overlap in observation distributions and frequent transitions among states leads to high mixing, thereby increasing the Dobrushin coefficient and ensuring filter stability.
>
> Additionally, Reviewer KcfL raised a similar concern. The reviewer may find the "Regarding Assumption 1" section of our rebuttal to Reviewer KcfL helpful, as it addresses this point in more detail.
>
> [1] Ali Devran Kara. 2022. Near optimality of finite memory feedback policies in partially observed Markov decision processes. J. Mach. Learn. Res. 23, 1, Article 11 (January 2022), 46 pages.
>
> ## Empirical Validation
>
> Reviewer TqLW has also made a similar comment. Due to space limitations, we have provided our response in the **Additional Experiments** section of the Comments to Reviewer TqLW, which provides an additional experiment comparing our work with Cayci et al.
>
> ## Writing Feedback and Additional Comments made by the Reviewer
>
> Thank you for pointing this out — we will address the formatting concerns in the revised draft

---

> > ### Comment · Reviewer_jAwy · 2025-08-04
> >
> > I thank the author for the detailed rebuttal. I read the other reviews and comments, and am still concerned about assumption 1, as mentioned by reviewer KcfL as it might not hold in many "classic" POMDP. I understand that it might be ok in some settings where the observation is for example a noisy reading of the state. The comparison table provided to reviewer TqLW, is helpful to support the results.
> >
> > The details provided clarified my concerns on the proof, I will update my score accordingly.

---

### Official Review · Reviewer_TqLW · 2025-06-24

**Clarity:** 3
**Significance:** 3
**Originality:** 3
**Rating:** 4
**Confidence:** 2

**Summary:**

This paper proposes a framework for approximately solving Partially Observable Markov Decision Processes (POMDPs) by transforming them into a fully observable finite-state Markov Decision Process (MDP) over superstates, truncated observation-action histories of fixed length. The key contributions are two-fold:
- Theoretical Improvement: The authors derive a novel algebraic inequality (Lemma 2) that leads to an improved bound on the difference between the optimal value function of the original POMDP and that of the superstate MDP. The new bound decays exponentially with the history length.
- Algorithmic Development: Building on this approximation, the paper proposes a policy optimization algorithm that alternates between TD learning and policy updates, and provides theoretical convergence guarantees, even under function approximation with linear features.
These results naturally support the success of RNN based Deep RL algorithms for POMDP.

**Questions:**

Given the connection to memory-based models like RNNs or Transformers, do you envision your theoretical results extending to neural function approximations, or at least guiding their design?

**Ethical Concerns:**

["NO or VERY MINOR ethics concerns only"]

**Final Justification:**

The main concern was the implementability of the method, since it is often the case that POMDP algorithm with theoretical guarantees is hard to implement. The additional experiments provided by the author resolved the concern.
I would maintain the score with accept recommendation.

**Limitations:**

As mentioned in the weaknesses, the paper lacks a discussion of its limitations.
Although the authors claim that limitations are addressed in the conclusion, I could not find any such discussion.
At a minimum, the paper should include a discussion on the practical limitations regarding the direct implementability of the proposed algorithm.

**Quality:**

3

**Strengths And Weaknesses:**

**Strengths**
- General Algebraic Lemma: The paper introduces a simple yet powerful algebraic inequality (Lemma 2), which can be useful beyond this particular setting. Its generality suggests potential applications to a broader class of approximation analyses in RL and beyond.
- Theoretical Justification for Practical Methods: The superstate MDP construction and the use of TD learning closely resemble many practical POMDP approaches, such as RNN-based methods and finite memory Q-learning (e.g., Recurrent DQN). The framework thus offers a theoretical explanation for the empirical success of such methods.

**Weaknesses**
- Lack of Empirical Validation: Although the theoretical results are sound and meaningful and the empirical validation may not be the scope of this paper, even a simple toy example could help ground the theory.
- Limited Discussion of Limitations: While the theoretical contributions are strong, the paper does not clearly discuss the limitations of the approach. For instance, how large must the history length be in practice to get a reasonable approximation.

---

> ### Author Rebuttal · Authors · 2025-07-30
>
> We sincerely thank the reviewer for their thoughtful comments and valuable feedback, which have helped us improve the quality and clarity of the paper. We hope that the following clarifications help address the reviewer's concerns.
>
> ## Additional Experiments
>
> Thank you for the comment. We do have a simple experiment in the Appendix, which shows the improvement in the performance as the length of the fixed truncated history is increased. We have now added one additional example which we will present later in our response. Additionally, we will add more experiments in the longer version of the paper. A common approach in the RL community is to heuristically use a history of observations as the state, thus converting a POMDP problem into an MDP problem. Our main goal in the paper was to theoretically prove that this approach yields good performance. Prior theoretical results indicated that the performance was much poorer than what we were able to show.
>
> For now, we have expanded on the simulations presented in the appendix of our paper in the following manner. In our paper, we show that we improve upon the bound in Cayci et al. using a computationally lighter algorithm. To experimentally illustrate this result, we consider an example for a simple partially observable Markov decision process (POMDP) with two states, two actions, and two observations. The environment is stochastic, with state transitions and observations defined by fixed probabilities, and rewards designed to encourage taking the correct action in the hidden state.
>
> To handle partial observability, we represent the agent’s state by a finite history of recent action-observation pairs, with history lengths of 1 and 2 tested.
>
> For each setting, the agent trains over 200 episodes, each of fixed length 20 steps. The learning rate $\alpha$ is set to 0.1 and the discount factor $\gamma$ to 0.9. Policies are represented tabularly and updated greedily with respect to Q-values after each episode.
>
> We measure the agent's performance by total reward accumulated per episode and analyze learning curves as well as final average rewards (averaged over the last 20 episodes) to assess convergence and policy quality. Since we are not allowed to share figures as a rebuttal, we are presenting a table of relevant values to demonstrate the improvement of our approach against Cayci et al.
>
> | History Length (H) | Algorithm       | Episodes | EpLen | Alpha | Gamma | AvgReward |
> |--------------------|------------------|----------|--------|--------|--------|------------|
> | 1                  | Our              | 200      | 20     | 0.10   | 0.90   | 14.100     |
> | 1                  | Cayci et al.     | 200      | 20     | 0.10   | 0.90   | 7.900      |
> | 2                  | Our              | 200      | 20     | 0.10   | 0.90   | 16.100     |
> | 2                  | Cayci et al.     | 200      | 20     | 0.10   | 0.90   | 14.900     |
>
>
> Additionally, since plots and figures are not allowed to be submitted for rebuttals, we present a short table which captures a moving average reward at episode numbers {20,40,60,80,100,120,140,160,180,200}
>
> | Episode | H=1 Our | H=1 Cayci et al. | H=2 Our | H=2 Cayci et al. |
> |---------|---------|-------------------|---------|-------------------|
> | 20      | 14.00   | 8.25              | 14.75   | 15.05             |
> | 40      | 14.00   | 7.70              | 15.35   | 13.30             |
> | 60      | 13.30   | 8.30              | 16.90   | 13.15             |
> | 80      | 13.20   | 8.20              | 16.15   | 14.75             |
> | 100     | 13.80   | 8.20              | 14.45   | 14.25             |
> | 120     | 13.70   | 7.60              | 16.20   | 12.90             |
> | 140     | 14.20   | 7.80              | 16.20   | 14.05             |
> | 160     | 14.30   | 7.70              | 14.90   | 14.10             |
> | 180     | 14.20   | 8.00              | 14.45   | 14.00             |
> | 200     | 14.25   | 7.95              | 16.40   | 13.30             |
>
>
> From this, we clearly see that 1) Increasing history length provides better results 2) Our algorithm performs better than Cayci et al. in terms of average reward.
>
> Additionally, several prior works have empirically demonstrated the effectiveness of using finite histories of observations as surrogates for hidden states in partially observable reinforcement learning problems. For example, [1] leverages pretrained language transformers to compress observation histories into compact representations, showing improved sample efficiency on POMDP benchmarks. This is similar to our work if we consider the feature vectors generated by the language transformers as the feature vectors. Similarly, [2] presents a framework that adaptively uses privileged state information during training while deploying policies that rely on observation histories. Other related studies, such as [3] and [4], support the use of history-based or memory-augmented policies with policy gradient methods.
>
> These empirical results suggest that the framework we analyze theoretically already has practical viability in reinforcement learning applications. However, we note that our work is the first to provide much tighter theoretical guarantees than available before.
>
> [1] Paischer, Fabian & Adler, Thomas & Patil, Vihang & Bitto-Nemling, Angela & Holzleitner, Markus & Lehner, Sebastian & Eghbalzadeh, Hamid & Hochreiter, Sepp. (2022). History Compression via Language Models in Reinforcement Learning. 10.48550/arXiv.2205.12258.
>
> [2] Jinqiu Li, Enmin Zhao, Tong Wei, Junliang Xing, and Shiming Xiang. 2025. Dual critic reinforcement learning under partial observability. In Proceedings of the 38th International Conference on Neural Information Processing Systems (NIPS '24), Vol. 37. Curran Associates Inc., Red Hook, NY, USA, Article 3704, 116676–116704.
>
> [3] Ni, T., Eysenbach, B. &amp; Salakhutdinov, R.. (2022). Recurrent Model-Free RL Can Be a Strong Baseline for Many POMDPs. Proceedings of the 39th International Conference on Machine Learning, in Proceedings of Machine Learning Research 162:16691-16723
>
> [4] Aberdeen, D., Buffet, O. &amp; Thomas, O.. (2007). Policy-Gradients for PSRs and POMDPs. Proceedings of the Eleventh International Conference on Artificial Intelligence and Statistics, in Proceedings of Machine Learning Research 2:3-10
>
> ## Discussion on Limitations
>
> The analysis is limited to discrete states and actions. Additionally, our analysis is limited to linear function approximation. Therefore, a potential future work includes extending the analysis to RNNs or Transformer models. We will explicitly mention these limitations in the revised draft.
>
>  ## Extension to RNNs or Neural function approximation
>
>  In our paper, we focused on the linear function approximation case; however, we believe that a proof technique inspired by the proof and assumptions made in Cayci and Eryilmaz [2024] can be used to extend our analysis to RNNs. Additionally, one may use ideas from Neural Tangent Kernel (NTK) theory to extend results to neural function approximations.

---

> > ### Comment · Reviewer_TqLW · 2025-08-02
> >
> > Dear authors,
> > Thank you very much for your time and efforts to address the concerns.
> > The additional experiment strengthen the implementability of the method and the limitations are clearly discussed.
> > I would maintain the score, recommending accept.

---

> > > ### Author Response · Authors · 2025-08-02
> > >
> > > Dear Reviewer TqLW, thank you for your comment to our response. We noticed that your score was borderline accept before the rebuttal.  Since your response says that you are recommending accept, can you confirm whether you are changing it to accept based on your comment? Thanks.

---

> > > > ### Comment · Reviewer_TqLW · 2025-08-04
> > > >
> > > > Thank you for your follow-up message. To clarify, my initial score was borderline accept, and after reading your rebuttal and additional experiments, I am maintaining the same score — which corresponds to a recommendation to accept. I hope this resolves any ambiguity.

---

### Official Review · Reviewer_KcfL · 2025-07-02

**Clarity:** 3
**Significance:** 2
**Originality:** 3
**Rating:** 4
**Confidence:** 4

**Summary:**

The paper presents an improved RL algorithm for POMDPs based on the idea of operating over so-called *superstates*, which are history suffixes.  In fact, optimal behaviour is possible by acting according to *belief states*, which are a function of the current history of past observations and actions, since a belief corresponds to the probability distribution over the hidden states given the current history.  However, using beliefs directly is not generally tractable, and many different alternatives have been proposed in the literature, relying on either assumptions or heuristics.
Superstates are an approach where histories are approximated by their suffixes of a given length $l$.
The paper proves approximation guarantees for superstates, in terms of how accurately the value functions of the original POMDPs are captured by the MDP induced by superstates. The approximation guarantees provide the foundation to develop tractable RL techniques.  Specifically, the authors show convergence guarantees for policy optimisation, as well as performance bounds in the case of linear function approximation.

**Questions:**

**Q1.** What is the intuitive meaning of Assumption 1?

**Q2.** How restrictive is Assumption 1? Could you please provide arguments on its generality, and provide examples where the assumption holds?

**Q3.** Could you please discuss some of the assumptions of the algorithms you improve on, discussing how such assumptions related to your Assumption 1?

**Q4.** Could you please provide an intuitive argument on why Lemma 1 holds? In other words, the high-level argument of the proof.

**Ethical Concerns:**

["NO or VERY MINOR ethics concerns only"]

**Final Justification:**

The discussion has helped me to get a better idea of the significance of the contribution. It is a solid paper overall, addressing a hard problem. However, the central Assumption 1 limits the applicability of the proposed techniques, and makes the contribution rather incremental compared to previous work. I will keep my score, which I believe it fairly summarises my assessment as just summarised.

**Limitations:**

The main limitation is the lack of a discussion on how general or restrictive the main assumption (Assumption 1) is.

The example I provide above shows the limitations on the applicability of the proposed techniques.

**Paper Formatting Concerns:**

I will provide a few remarks on notation.


Line 149, in the definition of $\Phi$, the two sides should be swapped, since this way it is as you are defining a probability in terms of the undefined function $\Phi$.

The symbol $\mathbf{\pi}$ is used for belief states. However, $\pi$ is the standard symbol for a policy. I suggest to replace it with a different symbol, e.g, I would use $\mathbf{b}$, hinting at beliefs.

The symbol $\mu$ is used for policies. However, the standard symbol for policies is $\pi$. Thus I suggest to use $\pi$ instead of $\mu$. Note that this relates to the comment above.

In Equation (3), the notation $H$ $\cup$ { $y,a$ } is used to denote that the pair $y,a$ is appended to the current history $H$. However, such set notation is not appropriate, and it should be improved. There are occurrences of such notation throughout the paper.

Assumption 1 uses $v$ for belief states, whereas in the rest of the paper they are denoted by $\mathbf{\pi}$. I suggest to use the same notation everywhere.

**Quality:**

3

**Strengths And Weaknesses:**

## Strengths

The research topic is **interesting and impactful**. RL algorithms are a challenging and active area of research. Improvements in these field are likely to have a significant impact on the research area. They may also have an impact in practice depending on applicability of the results. However, as it is the case for this paper, it is unclear how restrictive the required assumptions are in practice.

The paper looks **solid from a technical point of view**, except for some minor issues in the formal statements (e.g., some undefined notions, some glitches in notation).

The **presentation is clear**. The paper is easy to follow.


## Weakenesses


### The main assumption (Assumption 1)

The meaning (or intuition) of Assumption 1 is not given. It is only said that "this condition ensures sufficient mixing, preventing the system from being trapped in a subset of states".  Even more importantly, it is not discussed whether it is a reasonable assumption, e.g., how restrictive it is, and in which settings is expected to hold true.  Providing such a discussion is essential to substantiate the following claim in the conclusions:
"We show that standard policy optimization algorithms can effectively approximate an optimal POMDP policy ... without *restrictive assumptions* ..."

**Assumption 1 is clearly restrictive** in some way. In fact, I provide below a very simple POMDP for which the central Theorem 2 does not hold, implying that Assumption 1 is violated, unless the theorem is false. Thus, in order to support the significance of the contribution, it is extremely important to clarify in what way Assumption 1 is less restrictive than the assumptions considered in the existing literature.

### Example (T-maze)

The T-maze is an elementary domain introduced in [*Reinforcement learning with long short-term memory*, Bram Bakker, NeurIPS 2021]. Here we present an infinite-horizon variant. The domain consists of a T-shaped maze, sketched in the figure below. The agent is initially located at the beginning of a corridor (position $\mathtt{S}$), which is followed by a junction (position $\mathtt{X}$) where the agent can step into two different directions, and can keep going into that direction. The goal of the agent is to step into the correct direction, where the correct direction is determined by a piece of information the agent is given at the very beginning, when he is located at the beginning of the corridor (position $\mathtt{S}$).  The agent is given a non-zero reward $R > 0$ in every cell after taking the correct direction.

                        .
                        .
                        .
                        |
                        |
    S-------------------X
                        |
                        |
                        .
                        .
                        .

The T-maze is naturally modelled as a POMDP. The hidden states consist of
- an initial state $s_0$,
- states $s_i^d$ for every position $i$ in the corridor, including the junction, standing for the fact that the agent is in position $i$ and the initial observation specified that the correct direction is $d \in$ { $1,2$ } (say, e.g., that $1$ and $2$ stand for 'left' and 'right')
- states $r_i^d$ for $i \in \mathbb{N}$ and $d \in$ { $1, 2$ }, standing for the fact that the agent is in the $i$-th cell after stepping into the first of the two possible directions, and given that the initial observation said that the correct direction to take is direction $d$.
- states $q_i^d$ for $i \in \mathbb{N}$ and $d \in$ { $1, 2$ }, standing for the fact that the agent is in the $i$-th cell after stepping into the second of the two possible directions, and given that the initial observation said that the correct direction to take is direction $d$.


Note that transitions in the hidden state space are deterministic.  Given the entire history of observations and actions, the agent can determine the current hidden state, hence belief states are sharp, always assigning probability $1$ to a hidden state and $0$ to the others.  If only a suffix of the history is available, the agent can only determine its current location, and it cannot distinguish between states $s_i^1$ and $s_i^1$, between states $r_i^1$ and $r_i^2$, and between states $q_i^1$ and $q_i^2$.  In other words, belief states will assign probability $1/2$ to each of the two possible states.

The optimal value, given an entire history $H$ is roughly $V^* = R \cdot (1/1-\gamma)$.  However, the value of any superstate $G(H)$ is roughly $V^* / 2$, since the value is determined by the belief, which is uniform over non-distinguishable stats once we remove the first observation. The difference between the two values i  difference is $V^* / 2 = R/2 \cdot (1/1-\gamma)$, which is constant wrt to $l$, as opposed to the bound provided by Theorem 2, which converges to $0$ when $l \to \infty$.



### No experimental evalution

An experimental evaluation would help to assess the practical effectiveness of the proposed techniques. In particular, it would allow for assessing the impact of the main assumption (Assumption 1), and to draw conclusions on whether it is restrictive or not in terms of the existing benchmarks.



## Additional Comments


*Section 3.1.* The random variables $a_t$ for actions are never introduced. They should be introduced.

*Section 3.1.* Rewards are not introduced, and they are an essential part of POMDPs. As a consequence of this, Lines 167 and Equation (3) mention the function $r(s,a)$ -- reward function of hidden state and action -- which is undefined.

The definitions of *value function* and *Q-value function* need to be slightly improved. Currently, their definitions index the random variables $\mathbf{\pi}_t$ and $a_t$ starting from t = 0. Thus, the definition of value function is valid only for the case where the given belief state $\mathbf{\pi}$ is the initial belief state $\mathbf{\pi}_0$. Similarly, the definition of Q-value function is valid only for the case where the given belief state $\mathbf{\pi}$ and action $a$ are the initial belief state $\mathbf{\pi}_0$ and the initial action $a_0$. In fact, $s_t$ is defined as the state at time $t$ (Line 139), and $a_t$ is not defined but I assume it it the action at time $t$.

The definitions should be generalised to take into account the possibility that the given belief state $\mathbf{\pi}$ and action $a$ are the belief state $\mathbf{\pi}_t$ and action $a_t$ at time $t$. This is important at least from a conceptual point of view, because otherwise it is not made explicit that a state or state-action pair occurring at time $t$ have a value. The general definition can be obtained by parametrising the two value functions by the time-step variable $t$. Then, the current definitions are recovered as the special case when $t = 0$.

Assumption 1 mentions the quantity $K_{a,y}$, which is undefined.

---

> ### Author Rebuttal · Authors · 2025-07-30
>
> We sincerely thank the reviewer for their thoughtful comments and valuable feedback, which have helped us improve the quality and clarity of the paper. We hope that the following clarifications help address the reviewer's concerns.
>
> ## Regarding Assumption 1
>
> Assumption 1, i.e., the filter stability condition, means that every new observation sufficiently informs the agent to reduce differences between any two prior beliefs, ensuring the belief state “forgets” initial uncertainty and the filtering process is well-behaved. Therefore, one intuitive condition when Assumption 1 holds is when **the belief update is less sensitive to the prior and more influenced by new observations and actions**. This means that the next action and observation dictate what the next belief will be more than the current belief of the system.
>
> Therefore, to ensure that the filter stability condition holds, we need
>
> 1) **Sufficiently Mixing State Transitions**: The transition kernel $\mathcal{P}(s'\mid s,a)$ should be mixing, meaning from any state s, there's a positive probability to reach many other states s' over time.
>
> 2) **Non-deterministic and Informative Observations**: The observation kernel $\phi(y\mid s)$ must be non-deterministic but sufficiently informative.
>
> To check whether the filter stability condition holds, one sufficient condition uses the Dobrushin Coefficients of the Transition kernel and the Observation Kernel.
>
> From Theorem 5 of [1], if $(1-\delta(P))(2-\delta(\phi)) < 1$, then the filter stability condition holds. Here $\delta(\mathcal{P})$ is the minimum Dobrushin coeffient of the Transition kernel across all possible actions and $\delta(\phi)$ is the Dobrushin coefficient of the Observation kernel.
>
> Let P be a row-stochastic matrix over a finite state space $\mathbb{S}$. The Dobrushin coefficient $\delta(P)$ is defined as: $\delta(P(.\mid.,a)) = \inf_{x, y \in \mathbb{S}} \sum_{z \in \mathbb{S}} \min( P(z\mid x,a), P(z \mid y,a)) $
>
> and
> $\delta(P) = \min_a \delta(P(.\mid.,a))$
>
> This quantifies the minimum overlap between any two rows of the matrix P.
>
> This quantity measures how similar the rows of the matrix are; larger values indicate more mixing and hence, stronger contraction properties. As a result, when there is sufficient mixing—i.e., when every state has a non-trivial probability of transitioning to several other states or when every observation can be generated from multiple underlying states—the system exhibits filter stability. This scenario is common in dynamic environments and/or environments that have noisy observations, where the belief update is less sensitive to the prior and more influenced by new observations.
>
> A practical example is **Customer Behavior Modeling in Retail**. Here:
>
> States represent engagement levels such as {Uninterested, Browsing, Considering, Purchasing}.
>
> Observations are features like the number and type of clicks (e.g., “Viewed Product”, “Added to Cart”).
>
> In such systems, observations from adjacent engagement states tend to overlap significantly: for instance, both “Browsing” and “Considering” may involve product views and occasional cart additions. Furthermore, customer behavior is highly dynamic—users frequently move between these states in short time spans (e.g., from "Considering" back to "Browsing" or forward to "Purchasing"). This overlap in observation distributions and frequent transitions among states leads to high mixing, thereby increasing the Dobrushin coefficient and ensuring filter stability.
>
> A Simplified example to model customer behaviour is as follows:
>
> **States**: $s_0$ (Uninterested), $s_1$ (browsing), $s_2$ (Considering), $s_3$ (Purchasing).
>
> **Actions**: $a_0$: Show generic homepage, $a_1$: Recommend trending products
>
> **Observations**: $y_0$: No clicks, $y_1$: Viewed product, $y_2$: Added to cart, $y_3$: Purchased
>
> Suppose the transition probabilities for $a_0$ are
>
> | $s_t$ → $s_{t+1}$ | $s_0$ | $s_1$ | $s_2$ | $s_3$ |
> | ----------------- | ----- | ----- | ----- | ----- |
> | **$s_0$**         | 0.4   | 0.4   | 0.1   | 0.1   |
> | **$s_1$**         | 0.3   | 0.3   | 0.2   | 0.2   |
> | **$s_2$**         | 0.2   | 0.3   | 0.3   | 0.2   |
> | **$s_3$**         | 0.1   | 0.2   | 0.4   | 0.3   |
>
> and for $a_1$ be
>
> | $s_t$ → $s_{t+1}$ | $s_0$ | $s_1$ | $s_2$ | $s_3$ |
> | ----------------- | ----- | ----- | ----- | ----- |
> | **$s_0$**         | 0.4   | 0.3   | 0.2   | 0.1   |
> | **$s_1$**         | 0.2   | 0.4   | 0.2   | 0.2   |
> | **$s_2$**         | 0.1   | 0.3   | 0.4   | 0.2   |
> | **$s_3$**         | 0.1   | 0.2   | 0.3   | 0.4   |
> and the observation kernel is
>
> | $s_t$ → $y_t$ | $y_0$ | $y_1$ | $y_2$ | $y_3$ |
> | ------------- | ----- | ----- | ----- | ----- |
> | **$s_0$**     | 0.8   | 0.2   | 0.0   | 0.0   |
> | **$s_1$**     | 0.3   | 0.5   | 0.2   | 0.0   |
> | **$s_2$**     | 0.1   | 0.3   | 0.4   | 0.2   |
> | **$s_3$**     | 0.0   | 0.1   | 0.3   | 0.6   |
>
> Therefore, $\delta(P) = 0.5$ and $\delta(\phi) = 0.1$, which satisfies the sufficient condition $(1-\delta(P))(2-\delta(\phi)) < 1$, which implies that the Filter Stability condition is satisfied for this case.
>
> Additionally, for applications that do not satisfy Assumption 1, one could consider a multi-step variant where the system exhibits contraction after every k steps. We believe our results could be extended under this weaker assumption, making it a promising direction for future work.
>
>
> [1] Ali Devran Kara. 2022. Near optimality of finite memory feedback policies in partially observed Markov decision processes. J. Mach. Learn. Res. 23, 1, Article 11 (January 2022), 46 pages.
>
> ## Comparison with Other Works
>
> Cayci et al. consider an ergodicity condition that requires every superstate to be visited within a finite time interval. Under this assumption, they establish a filter stability condition similar to Assumption 1 in our work. However, to achieve meaningful performance guarantees, their method relies on an m-step TD learning setup to obtain accurate estimates of the Q-values, which significantly increases the computational cost of their algorithm. In contrast, our approach yields improved theoretical bounds with lower computational overhead.
>
> Kara et al. adopt the same one-step filter stability condition used in our paper. This assumption is a standard tool in the analysis of partially observed systems, widely employed in control theory, filtering, and the POMDP literature to establish convergence, approximation, and learning guarantees. It provides a tractable framework for analyzing the evolution and stability of belief states over time. However, we note that Kara et al. only considered Q-learning without function approximation. One of our key contributions is to show that the same assumption allows us to get finite-time performance bounds for Policy Optimization using new techniques.
>
> In comparison, several other works adopt the much stronger n-step decodability assumption, which assumes the cumulative observation sequence over n steps becomes informative enough to infer the underlying state. We believe that this is a significantly more restrictive condition than the filter stability assumption used in our analysis.
>
> ## Intuition about Lemma 1
>
> If two histories belong to the same superstate, then their sequences of (action, observation) pairs over the past l timesteps are identical. The key intuition behind Lemma 1 is that, when observations are sufficiently informative, the recent (action, observation) history contains enough information to accurately infer the current system state. As a result, starting from two different initial belief states, the combination of informative observations and strong mixing guarantees that, after a sufficiently long time, the corresponding belief states will converge to similar distributions.
>
> ## Regarding More Experimental Evaluations
>
> Reviewer TqLW has also made a similar comment. Due to space limitations, we have provided our response in the **Additional Experiments** section of the Comments to Reviewer TqLW.
>
> ## Paper Formatting Concerns and Additional Comments made by the Reviewer
>
> Thank you for pointing this out — we will address the formatting concerns in the revised draft

---

> > ### Comment · Reviewer_KcfL · 2025-08-06
> >
> > I would like to thank the authors for their response. It clarifies my doubts. I encourage the authors to improve the paper by including the contents of their response.
> >
> > Regarding Assumption 1, I find it rather restrictive. At the same time, I acknowledge how hard the considered problem is, and that the results of this paper are an improvement over the state of the art. I encourage the authors to clearly describe the limitations imposed by Assumption 1. In my opinion, including an example along the lines of my example would make it clear what kind of domains are excluded by the assumption.

---

> > > ### Author Response · Authors · 2025-08-06
> > >
> > > We thank the reviewer for their valuable suggestions. We would be happy to include a discussion on the types of domains where Assumption 1 is valid. We also appreciate the reviewer’s counterexample and would be glad to include it in the paper.

---

### Comment · Area_Chair_YDMG · 2025-08-04
**Please check out the author rebuttals**

If you have not already done so, please check out the authors rebuttals. There is a small window left to engage with the authors and seek additional clarifications if needed.

---

### Note · Authors · 2025-08-13

We thank all the reviewers for their valuable comments. Their feedback has been helpful in further improving our paper. We will make sure that we incorporate the reviewers’ suggestions into the final version.

---

### Decision · Program_Chairs · 2025-09-17

**Decision:**

Accept (poster)

**Comment:**

I am weakly positive on this paper. It contains interesting theoretical results that deserve an audience. However, it makes an assumption (assumption 1) that may be reasonable but that is not, as far as I can tell, widely adopted and known - unless it goes by another name. This assumption was not explained and justified in a way that made the reviewers comfortable that it was a clear and reasonable assumption. The paper did not initially provide any compelling experiments or detailed discussions of practical domains that might reassure readers that the assumptions were reasonable.

After some back and forth with the reviewers, the authors did address some of these concerns. They explained assumption 1 better, and they provided some modest experiments. This was sufficient to move all of the reviewers into "borderline accept", but none were interested is championing the paper. (In discussion, I asked if anybody was willing to champion the paper, and no reviewers did so.)

In summary, I think that this is interesting work that - as initially submitted - was less compelling than it could be due to minimally discussed assumptions and lack of experiments. The authors demonstrated that they can make a more compelling case, but the rebuttal process got them just barely over the bar in terms of being convincing to reviewers. I think this paper could be a fine poster for NeurIPS, though I also wonder if the authors could have more impact overall if they submitted to a future venue and included more compelling experiments and justifications in that paper.